# Anti-Retroviral Protease Inhibitors Regulate Human Papillomavirus 16 Infection of Primary Oral and Cervical Epithelium

**DOI:** 10.3390/cancers12092664

**Published:** 2020-09-18

**Authors:** Samina Alam, Sreejata Chatterjee, Sa Do Kang, Janice Milici, Jennifer Biryukov, Han Chen, Craig Meyers

**Affiliations:** 1Department of Microbiology and Immunology, The Pennsylvania State University College of Medicine, 500 University Drive, Hershey, PA 17033, USA; salam@pennstatehealth.psu.edu (S.A.); sreejatach@gmail.com (S.C.); skang@pennstatehealth.psu.edu (S.D.K.); jmilici@pennstatehealth.psu.edu (J.M.); jenbiryukov@gmail.com (J.B.); 2Section of Research Resources, The Pennsylvania State University College of Medicine, Hershey, PA 17033, USA; hchen3@pennstatehealth.psu.edu; 3Department of Obstetrics and Gynecology, The Pennsylvania State University College of Medicine, Hershey, PA 17033, USA

**Keywords:** non-AIDS defining cancers (NADC), AIDS defining cancers (ADC), human papillomavirus type 16 (HPV16), opportunistic HPV infections, oropharyngeal cancer, cervical cancer, highly active anti-retroviral therapy (HAART), protease inhibitors, organotypic raft cultures, HPV persistence

## Abstract

**Simple Summary:**

In 2016, globally, 36.7 million people were living with Human Immunodeficiency Virus (HIV), of which 53% had access to anti-retroviral therapy (ART) (UNAIDS 2017 Global HIV Statistics). The risk of Human Papillomavirus (HPV) associated oropharyngeal, cervical and anal cancers are higher among patients infected with HIV in the era of ART. Generally, HPV infections are self-limiting, however, persistent HPV infection is a major risk to carcinogenic progression. Long intervals between initial infection and cancer development imply cofactors are involved. Co-factors that increase infectivity, viral load, and persistence increase risk of cancer. We propose that the ART Protease Inhibitors (PI) class of drugs are novel co-factors that regulate HPV infection in HIV-infected patients. We developed a model system of organotypic epithelium to study impact of PI treatment on HPV16 infection. Our model could be used to study mechanisms of HPV infection in context of ART, and for developing drugs that minimize HPV infections.

**Abstract:**

Epidemiology studies suggest that Human Immunodeficiency Virus (HIV)-infected patients on highly active anti-retroviral therapy (HAART) may be at increased risk of acquiring opportunistic Human Papillomavirus (HPV) infections and developing oral and cervical cancers. Effective HAART usage has improved survival but increased the risk for HPV-associated cancers. In this manuscript, we report that Protease Inhibitors (PI) treatment of three-dimensional tissues derived from primary human gingiva and cervical epithelial cells compromised cell-cell junctions within stratified epithelium and enhanced paracellular permeability of HPV16 to the basal layer for infection, culminating in de novo biosynthesis of progeny HPV16 as determined using 5-Bromo-2′-deoxyuridine (BrdU) labeling of newly synthesized genomes. We propose that HAART/PI represent a novel class of co-factors that modulate HPV infection of the target epithelium. Our in vitro tissue culture model is an important tool to study the mechanistic role of anti-retroviral drugs in promoting HPV infections in HAART-naïve primary epithelium. Changes in subsequent viral load could promote new infections, create HPV reservoirs that increase virus persistence, and increase the risk of oral and cervical cancer development in HIV-positive patients undergoing long-term HAART treatment.

## 1. Introduction

In 2016, globally, 36.7 million people were living with Human Immunodeficiency Virus (HIV), of which 53% had access to anti-retroviral therapy (ART) (UNAIDS 2017 Global HIV Statistics). In the United States, approximately 1.2 million people are living with Human Immunodeficiency Virus (HIV)/AIDS [1]. Better tolerated combinations of Highly Active Anti-Retroviral Therapy (HAART) has significantly improved the survival of HIV-positive individuals including reduction of new infections and extended life-span [2]. Declined mortality among HIV-infected individuals has resulted in growth and aging of the HIV-positive populations, which has implications for increased risk of cancer development [3]. Higher cancer risk in HIV/AIDS patients compared to the general population is a result of HIV-related immunosuppression that impairs control of oncogenic viral infections [3], including AIDS-defining cancers (ADC) due to HPV induced cervical cancer, and non-AIDS-defining cancers (NADC) that include HPV associated oropharyngeal and anal cancers [3,4]. Since the introduction of HAART in 1996, rates of ADC, including cervical cancer have decreased, but incident rates of cervical cancer remain elevated in patients undergoing HAART treatment compared to the general population [3]. In contrast, rates of NADCs have increased with respect to both oropharyngeal and anal cancers [3]. Although increasing longevity is the greatest risk factor for NADCs, it is insufficient to explain trends in cancer epidemiology [5].

In 1995, the first generation of Protease Inhibitors (PI) class of anti-retroviral therapy (ART) became commercially available, followed by novel combinations of PIs with other anti-retroviral classes of drugs [5,6]. Multiple epidemiological studies representing diverse cohorts support the finding that oral manifestations of HPV infections increased among HIV-positive patients on long-term HAART compared to patients not taking ART [7,8,9,10,11]. One study analyzed cancer incidence after ART initiation in eight US HIV clinical cohorts who started ART between 1996 and 2011, of which 50% started a PI-containing regimen [6], showed that rates of NADCs rose with longer time on ART [12], and older age was a significant predictor of NADCs, including HPV related malignancies [12]. A retrospective study of patients (1996–1999), analyzed the relationship between exposure to combination HAART therapy and prevalence of oral warts and showed that oral lesions were significantly associated with PI containing regimens compared with another class of HAART [8]. Prevalence of oral warts was 23% of patients on HAART (+PI) and 15% of patients on HAART (−PI) therapy containing HIV non-nucleoside reverse transcriptase inhibitors (NNRTI), when compared to 5% of patients on neither medication [8]. When adjusted for CD4+ count and HIV load, the odds of having oral warts for those on HAART + NNRTI alone showed a non-significant association, but for those on HAART + PI there was a highly significant association, which also suggested that HAART use increased oral warts [8]. Overall, HAART usage has decreased incidents of oral lesions of both viral- and non-viral etiologies and correlates with increased CD4+ T-cell count, but is not statistically significant for decreased HPV infections [13]. Therefore, the burden of NADCs continues to rise, as does the need for cancer detection, prevention and treatment in HIV-positive patients [14].

Studies suggest that patients on HAART are at increased risk of acquiring opportunistic HPV infections and developing oropharyngeal [15,16], anal [17] and cervical cancers [18], compared to the general population. Oncogenic high-risk HPV16 is responsible for more than 60% of oropharyngeal carcinoma, ~90% of tonsillar carcinomas [19], most cases of vulvar carcinoma [20], 90% of anal carcinoma [21] and penile cancers (less than 1% of all male cancers in the United States) with incidence rates that greatly vary across different regions of the world [21]. The oropharyngeal compartment is central to the persistence of HPV, and the virus is more commonly detected in the oral mucosa of HIV-positive patients compared to HIV-negative patients [22]. Up to 56% of HIV-infected adults have detectable HPV DNA, which is significantly higher than the non-HIV infected population [15,23]. In addition, HAART usage is associated with development of adverse oral complications that damage the mucosal epithelium and potentially expose the underlying tissues to HPV infection [24,25,26]. Thus, viral persistence could determine increased prevalence of oropharyngeal cancer among HIV patients receiving ART compared to the general population [8,16]. HPV16 tends to be persistent and is refractory to clearance in women on HAART [27], and accounts for ~70% of all invasive cervical cancer [28]. Cumulatively, these studies suggest that HAART treatment potentially enhances opportunistic HPV infection, viral persistence and cancer progression. 

The molecular mechanism of how HAART exposure sensitizes target epithelium to opportunistic HPV infection, and potentially other viruses, is of significant interest. In the current manuscript, we developed an in vitro model of HPV infection of HAART-naïve primary human gingiva and cervical epithelium, and asked whether treatment with two protease inhibitors, Amprenavir and Kaletra, prime the mucosa for virus infection, and further investigated impact of drug treatments on subsequent viral load. 

## 2. Results

### 2.1. Amprenavir Treatment Enhances HPV16 Infection of Primary Oral Tissue

Amprenavir (Agenerase^®^, GSK) was one of the first PIs in the market that was later removed due to increased viral resistance. The drug binds to the active site of HIV-1 aspartyl protease and prevents processing of viral gag and gag-pol poly-protein precursors resulting in formation of immature non-infectious viral particles [29]. Additionally, prolonged use of Amprenavir was associated with adverse orofacial effects including progressive oral warts that recurred after removal [8]. We previously reported that treatment with Amprenavir impacted growth, differentiation and epithelial repair of gingiva tissues [30]. In the current study, we determined whether PI exposure affected HPV16 infection in an in vitro model of three-dimensional epithelium. Organotypic cultures were derived from primary gingival keratinocytes isolated from mixed pools of human gingiva from patients undergoing dental surgery [30]. On day 8 post-lifting and differentiation at the air-liquid interface, gingiva tissues were treated for 24 h with 7.66 µg/mL Amprenavir (drug C_max_ representing peak blood concentration after drug administration maintained between two dosages for optimal HIV suppression). Drug treatment impacted tissue morphology, compromising cell-cell junctions within the stratified suprabasal as well as the basal layers, altering structural/barrier integrity of desmosome-, tight- and adherens junctions (Figure 1). It is generally thought that HPV infects cells of the basal layer via micro-abrasions, where viral genome amplifies to high copies [31]. Hypothetically, Amprenavir regulated damage of protein complexes at cell-cell contact sites is reminiscent of tissue “wounding” that could provide opportunistic HPV access to basal cells for infection. 

We then tested the impact of Amprenavir treatment on HPV16 infection of primary gingiva tissues. Laboratory stocks of HPV16 were prepared from raft tissues derived from cervical cell lines productively infected with HPV16 that were differentiated in culture for 20 d, as described in the Methods sections. Amprenavir concentrations ranging from 7.66–2.5 µg/mL were added to the culture media for 24–72 h, followed by infecting tissues at each time point using increasing doses of HPV16 virus particles as described in this scheme (Figure 2). As controls, untreated tissues were infected with the highest dose of HPV16 virions. Total time spent in culture was 15–18 d, respectively, followed by tissue harvesting and measuring the E1^E4 major spliced transcript, a major hallmark of virus infection. The E1^E4 open reading frame is present in both early and late HPV transcripts, and high level expression of this protein is restricted to differentiated suprabasal cells [32]. The E1^E4 protein has multiple functions and is thought to interact with keratin intermediate filament networks to facilitate network re-organization [33], associate with mitochondria to induce apoptosis [34], bind RNA processing proteins [35], disrupt nuclear dot 10 domains [36] and associate with cellular cyclins to mediate cell cycle arrest in the G1/M phase [37]. In addition, we have reported that the E1^E4 protein may also play a role in HPV capsid assembly, infectivity and virion maturation [38]. In the current manuscript, relative E1^E4 transcript levels in HPV16 infected/Amprenavir treated tissues were compared to HPV16 infected tissues that did not receive drug treatment. Amprenavir treatment renders primary gingiva tissue more favorable to HPV16 infection, compared to untreated tissues that were poorly infected (Figure 3A). Relative fold-change of E1^E4 expression varied relative to drug pre-treatment times across three independent experiments, and is attributed to natural variation in host cell genetics (Figure 3A and Appendix A). However, in all cases, E1^E4 transcript expression in drug treated tissues was significantly increased compared to drug untreated tissues. Throughout the manuscript, unless otherwise stated, for each experiment performed in triplicate, the first data panel is presented in the Results section, and the other two panels in the Appendix A section. Virus infection of tissue was inhibited using antibodies against HPV16 L1 (α-V5) and L2 (α-RG-1) capsid proteins (Figure 3B and Appendix A), thus demonstrating interference with receptor mediated virus entry pathways, as would be expected. Lower doses of Amprenavir (5 µg/mL) also significantly enhanced HPV16 infection only after 24–48 h of treatment (Figure 4A), whereas treatment with 2.5 µg/mL Amprenavir poorly supported infection (Figure 4B). These results are significant as we show for the first time that three-dimensional tissues derived from primary human epithelial cells can be infected with a high-risk HPV in vitro. Importantly, this could relate to the clinically relevant oral HPV16 infections observed in HIV+ patients undergoing HAART treatment. 

Further analysis showed that virus infection of Amprenavir (7.66 µg/mL) treated tissues correlated with changes in putative progeny viral titers, a milestone in the viral life-cycle (Figure 5, top panel and Appendix A). Such progeny HPV16 virions (*prog*-HPV16) poorly infected monolayer HaCaT cells compared to parental HPV16 (*P*-HPV16) used above for infecting raft tissues (Figure 5, bottom panel), suggesting that extended time in culture may be needed for virus capsid maturation to occur. This concept is based on our previously reported finding that improved HPV16 infectivity over time is a function of capsid maturation with respect to disulfide bond formation that determines virus infectivity [39,40]. Raft tissues treated with a range of Amprenavir concentrations (7.66, 5 and 2.5 µg/mL) and infected with either 7.5 × 10^7^ and 1.5 × 10^8^ of *P*-HPV16 virions, were cultured for a further 18–24 d, followed by preparing crude virus (CV) stocks of *prog*-HPV16 from tissues and virus titer determination (Figure 6A and Appendix A). In this experiment, Amprenavir dose-dependent changes in progeny virus titers over time was observed. Progeny HPV16 virus stocks isolated from tissues treated with 7.66 µg/mL Amprenavir were further purified using Optiprep gradient fractionation, where infectious virus particles with fully mature capsids are known to partition within fractions #5–#8 [41]. Infectivity of *prog*-HPV16 in fraction #7 [F#7] samples was measured by infecting HaCaT monolayer cultures, and determining fold-change expression of E1^E4 transcripts compared with *P*-HPV16 [F#7] virus (Figure 6B). Infectivity of *prog*-HPV16 [F#7] improved with increased time in culture and trended towards infectivity of *P*-HPV16 [F#7] stocks. Similar to *P*-HPV16, infection of *prog*-HPV16 [F#7] was inhibited with α-V5 and α-RG1 antibodies, suggesting that capsid structure was conserved between *prog*-HPV16 and *P*-HPV16 (Figure 6B). Virus infected gingiva tissues treated with 5 µg/mL Amprenavir also produced infectious *prog*-HPV16, that was neutralized with anti-capsid antibodies (Figure 7), suggesting that fluctuations of HAART concentrations as would occur in patients undergoing treatment, would not affect biochemical integrity of new HPV particles synthesized in target tissues. In contrast, long-term cultures treated with 2.5 µg/mL Amprenavir produced low titers of *prog*-HPV16 (Figure 6A and Appendix A). Cumulatively, these results suggest that Amprenavir acts as a co-factor to sensitize HPV16 infection and further increases viral load in oral tissues, at least via *prog*-HPV16 biosynthesis.

### 2.2. Using BrdU to Label Newly Synthesized HPV16 Genomes Distinguishes between Parental and Progeny Virions

To confirm that Amprenavir treatment induced de novo virus biosynthesis in gingiva tissues, we developed protocols to distinguish “input” *P*-HPV16 from “output” *prog*-HPV16 using BrdU labeling of newly replicated genomes. First, culture conditions were optimized for BrdU-labeling of replicating viral genomes, utilizing HPV16 positive organotypic cultures routinely used for generating infectious HPV16 standard laboratory stocks. In this system, high-risk HPV-positive cervical cell lines maintaining episomal copies of viral genomes are allowed to grow and differentiate over a period of 20 d [41], during which time tissue stratification occurs synonymously with viral genome amplification, late gene L1 and L2 capsid protein expression, culminating in virion morphogenesis [31]. We reported that HPV genome amplification occurs on day 8 prior to capsid protein expression on day 10, potentially in mechanistic tandem for genome encapsidation and virus assembly [42]. To maximally enable BrdU incorporation into replicating viral genomes with minimal toxicity to host tissues, on day 8 post-lifting of rafts to the air-liquid interface, 25 μM BrdU was added to media and maintained until tissue harvesting (Figure 8A). Infectivity of HPV16 stocks grown in the presence of BrdU (*P*-HPV16-BrdU) was similar to control/unlabeled *P*-HPV16, including the ability of neutralizing anti-capsid antibodies to inhibit virus infection of HaCaT monolayer cultures (Figure 8B).

To visualize HPV16-BrdU labeled genomes, HaCaT monolayer cultures were infected with *P*-HPV16-BrdU virus from [F#7]-stocks purified using Optiprep gradient fractionation, and immunofluorescence staining was performed to detect labeled genomes using a mouse anti-BrdU antibody. Additionally, to confirm that BrdU-labeled genomes were virion encapsidated, we further co-localized BrdU immunofluorescence with HPV16 L1 capsid protein using a rabbit anti-HPV16 L1 antibody. Significant co-localization was determined between viral genome/capsid complexes manifesting as punctate spots in perinuclear regions (Pearson’s correlation coefficient for co-localization between 0.560 ± 0.023 and 0.626 ± 0.076) (Figure 9A). Time course experiments were also performed to determine kinetics of HPV16-BrdU cell uptake and entry. The L1 capsid staining alone was weakly visualized at 3 h post-infection, suggesting virus attachment to cells (Figure 9B). However, BrdU immunofluorescence was not detected suggesting the absence of capsid uncoating. In contrast, at 5 h post-infection, significant co-localization was found between BrdU-labeled genomes and L1, suggesting initial stages of virus disassembly. Additionally, HPV16-BrdU staining appeared in polarized clusters, indicating mass virus trafficking within vesicles around perinuclear locations. In contrast, at 6 h post-infection, co-localization between L1 and genomes was decreased (Pearson’s coefficient 0.264 ± 0.017), indicating further advancement of viral capsid/genome disassembly. At 12 h post-infection, co-localization of the two fluorophores was not significant (Pearson’s coefficient 0.057 ± 0.019). As controls, non-specific binding of anti-BrdU and anti-L1 antibodies was not detected in uninfected HaCaT cells (Appendix A). Lastly, neutralizing α-V5/L1 capsid protein complexes were visualized at 6h post-infection, where L1 staining appeared in clusters depicting non-uniform spots (Figure 9B, right panel, compare L1 staining pattern with left panel, 6h post-infection without α-V5), that correlate with inhibition of virus infection (Figure 6B). Antibody sequestration or “clumping” of L1 presumably serves as a mechanism that prevents virus infectivity. Our studies present first time data that show biosynthesis of a high-risk papillomavirus in three-dimensional culture, that is amenable to BrdU-labeling and is able to be visualized following infection using confocal imaging.

### 2.3. BrdU Labels Newly Synthesized Progeny Virus in Amprenavir Treated Gingiva Tissue Infected with P-HPV16

The techniques developed above to label newly synthesized HPV16 genomes packaged into virions provided a foundation for further determining whether Amprenavir treated primary gingiva tissues infected with *P*-HPV16 also induced de novo *prog*-HPV16 biosynthesis. Primary tissues were optimized for BrdU-labeling of replicating genomes incorporated into new virions using a modified protocol (Figure 10). On day 8 post-lifting, primary gingiva tissues were pre-treated with Amprenavir for 72 h, followed by infecting with one of two doses of *P*-HPV16 as indicated. At 24 h post-infection, BrdU was added to the media and tissues were cultured until harvesting on day 22. Then, BrdU-labeled *prog*-HPV16 (*prog*-HPV16-BrdU) CV stocks were purified using Optiprep gradient fractionation. Monolayer HaCaT cultures were infected with [F#7]-derived *prog*-HPV16-BrdU (1 MOI) and visualized using confocal imaging (Figure 11A). Significant viral genome/L1 co-localization was determined 5 h post-infection (Pearson’s coefficient 0.657 ± 0.060), that was decreased at 6 h (Pearson’s coefficient 0.204 ± 0.109), suggesting that kinetics of virus disassembly was similar to *P*-HPV16-BrdU (Figure 9B). Further, punctate α-V5/L1 capsid-protein complexes were visualized that correlate with inhibition of infection (Figure 11A, right panel, compare L1 staining pattern with left panel, 6 h post-infection without α-V5; and Figure 6B). We present first time data that show Amprenavir treatment and infection of three-dimensional tissue with authentic HPV16 virus results in productive infection and de novo biosynthesis of infectious progeny HPV16 in vitro.

### 2.4. Kaletra^®^ Treatment Also Induces De Novo Biosynthesis of Progeny Virions in Gingiva Tissues

To confirm that our findings were not restricted to Amprenavir treatment, we utilized Kaletra (Lopinavir/ritonavir, Abbott Laboratories), a PI currently prescribed for HIV treatment. Use of Kaletra was approved in 2000 by the US FDA for the treatment of HIV infection in adults and children. Similar to Amprenavir, prolonged use of Kaletra is associated with several adverse orofacial effects [43,44]. We reported that Kaletra treatment also affected growth, differentiation and epithelial repair of gingiva tissues [45]. In the current study, we determined the impact of Kaletra treatment (C_max_ 9.8 µg/mL) on *P*-HPV16 infection and subsequent progeny viral load in primary tissues. Like Amprenavir, Kaletra treatment also rendered primary gingiva tissue more favorable to HPV16 infection, compared with untreated tissues (Figure 12A). Again, relative fold-change of E1^E4 expression varied relative to drug pre-treatment times across three independent experiments attributed to natural variation in host cell genetics (Figure 12A and Appendix A). Virus infection of tissue was neutralized using antibodies against HPV16 L1 and L2 capsid proteins (Figure 12B and Appendix A). Raft tissues treated with a range of Kaletra concentrations (9.8, 6 and 3 µg/mL) were infected with one of two doses of *P*-HPV16 and grown in long-term cultures, and putative *prog*-HPV16 CV stocks were titered and were further determined to be drug dose-dependent (Figure 13A and Appendix A). Infectivity of *prog*-HPV16 in [F#7]-stocks were similar to that of *P*-HPV16 [F#7], and were inhibited with α-V5 and α-RG1 antibodies, suggesting that regardless of whether Amprenavir or Kaletra was tested, capsid structure in relation to infectivity was conserved between *prog*-HPV16 and *P*-HPV16 (Figure 13B). In contrast, long-term cultures treated with 6 and 3 µg/mL Kaletra produced low *prog*-HPV16 titers (Figure 13A and Appendix A). We also confirmed that Kaletra treatment induced de novo virion biosynthesis. Our studies show that Kaletra treatment and infection of three-dimensional tissue with *P*-HPV16 virus also results in production of BrdU-labeled *prog*-HPV16 in vitro (Figure 11B). Time-line experiments were performed to show significant *prog*-HPV16 BrdU-labeled genome/L1 co-localization after 5 h of infection of HaCaT monolayer cells (Pearson’s coefficient 0.554 ± 0.069), that was decreased at 6 h post-infection (Pearson’s coefficient 0.172 ± 0.054) (Figure 11B). In addition, α-V5/L1 capsid protein complexes were similarly visualized that correlate with inhibition of infection (Figure 11B right panel, compare L1 staining pattern with left panel, 6 h post-infection without α-V5; and Figure 13B). We previously reported that Kaletra treatment of primary gingiva tissues also mediated disruption of protein complexes that regulate cell-cell junctions [45] that could provide HPVs with efficient access to their target cells in the basal layer. Overall, these results indicate that Kaletra also acts as a co-factor to sensitize HPV16 infection of oral tissue.

### 2.5. Amprenavir Treatment Allows for Virus Transit through Gingiva Tissue Layers for Infecting Basal Cells

Thus far, we have shown that untreated gingiva tissues were poorly infected with HPV16 compared with tissues treated with PI (Figure 3A and Figure 12A). Amprenavir mediated disruption of protein complexes that regulate cell-cell junctions (Figure 1) could provide HPVs with more efficient access to their target cells in the basal layer. In order to visually correlate HPV16 infection with virus localization within different layers of the stratified epithelium, we used *P*-HPV16-BrdU to infect Amprenavir treated primary gingiva tissues and performed immunofluorescent staining of BrdU-labeled HPV16 virus particles in transit through the tissue over a period of 12–72 h post-layering of virus on top of tissues (Figure 14A,B). At 12 h post-infection, HPV16-BrdU was mostly localized in the cornified and upper portions of the suprabasal layer irrespective of Amprenavir treatment. At increasing times post-addition of virus, in drug-treated tissues, HPV16-BrdU was localized throughout the suprabasal layer as well as cells within the basal layer, whereas in untreated tissues, the virus was restricted in the upper cornified layers of the tissues. Significant co-localization was determined between viral genome/capsid complexes, manifesting as punctate spots in perinuclear regions in cells of the basal layer (Pearson’s correlation coefficient for co-localization between 0.878 ± 0.094 and 0.644 ± 0.107) across 24–72 h post-layering of virus on tissues (Figure 14B). We also performed control experiments to rule out the possibility of “bleed-through” of the fluorescence emission of Alexa Fluor 488 and Alexa Fluor 568, the two fluorophores used in our studies, and found no evidence of crossover fluorescence (Appendix A). Taken together, our results suggest that Amprenavir mediated disruption of cell–cell barrier integrity likely plays a role in enhancing HPV16 transit through the epithelium to the target basal cells for infection, whereas non-disrupted cell-cell junctions in control tissues impede virus transit through the tissue layers.

### 2.6. Primary Cervix Tissues Differentially Regulate Progeny HPV16 Biosynthesis When Treated with Amprenavir and Kaletra

We also compared PI treatment and HPV16 infection of primary cervical epithelium and measured impact on de novo *prog*-HPV16 biosynthesis. Organotypic cultures were derived from primary cervical keratinocytes isolated from patients undergoing hysterectomy. Amprenavir and Kaletra were added at the C_max_ dosage (7.66 µg/mL and 9.8 µg/mL, respectively) to the culture media for 24–72 h, followed by infecting tissues using increasing doses of *P*-HPV16 (Figure 15A and Figure 16A). Untreated tissues infected with the highest dose of HPV16 virions were used as controls. Expression of the E1^E4 spliced transcript in drug treated tissues was measured and relative levels compared to HPV16 infected, drug untreated controls. Similar to gingiva tissues, treatment with Amprenavir and Kaletra, rendered primary cervical tissue more favorable to HPV16 infection, compared to untreated tissues that were poorly infected (Figure 15A and Appendix A; Figure 16A and Appendix A). Again, relative E1^E4 expression levels were non-linear with regard to drug pre-treatment times. Observed fold-changes of E1^E4 expression varied relative to drug pre-treatment times across three independent experiments, once again suggesting a universal role for host genetics of target tissues. Virus infected Amprenavir and Kaletra treated tissues, respectively, were also inhibited using antibodies against HPV16 L1 and L2 capsid proteins (Figure 15B and Appendix A; Figure 16B and Appendix A), as would be expected. These results are significant as we show for the first time that three-dimensional tissues derived from primary cervical epithelial cells can be infected with a high-risk HPV in vitro, in the context of HIV-positive patients co-infected with HPVs undergoing HAART treatment.

Further analysis showed that *P*-HPV16 infected tissues differentially correlated with changes in *prog*-HPV16 titers in the context of PI treatment utilized. Primary cervical tissues treated with Amprenavir (Figure 17A and Appendix A) induced virus titers that were comparable to those of infected gingiva tissues at all concentrations tested (Figure 6A and Appendix A). In contrast, treatment with the C_max_ dose of Kaletra negatively regulated progeny virus biosynthesis in cervical tissues (Figure 18A and Appendix A) compared with gingiva tissues (Figure 13A and Appendix A). Inability to synthesize *prog*-HPV16 in presence of Kaletra treatment was not due to ability of the cervical tissues to be infected, as clear infection, as determined using E1^E4 expression, was observed at all time points post-infection (Figure 16A and Appendix A). In contrast, treating cervical tissues with low concentrations of Kaletra (6–3 µg/mL) resulted in production of low yet measurable *prog*-HPV16 titers in long-term cultures (Figure 18A and Appendix A). Infectivity of *prog*-HPV16 produced in Amprenavir (7.66 µg/mL) or Kaletra (3 µg/mL) treated cervical tissue was similar to that of *P*-HPV16, as was the ability to be inhibited with α-V5 and α-RG1 antibodies, thereby suggesting that biochemical nature of newly synthesized progeny virions was conserved regardless of whether oral or cervical tissue was examined, or specific PI and concentration used for treatments (Figure 17B and Figure 18B). Progeny HPV16 biosynthesized in the presence of the two PIs was also confirmed using BrdU-labeling of genomes (Figure 11C,D). At 5 h post-infection in HaCaT monolayers, significant co-localization was also observed between BrdU labeled genomes and L1 capsids (Amprenavir 0.647 ± 0.050; Kaletra 0.782 ± 0.008), that was decreased at 6 h post-infection (Amprenavir 0.205 ± 0.098; Kaletra 0.208 ± 0.053), a further measure of capsid uncoating within infected cells. Additionally, α-V5/L1 capsid protein complexes correlate with inhibited infection (Figure 11C,D, right panels, compare L1 staining pattern with left panel, 6 h post-infection without α-V5; and Figure 17B and Figure 18B). Cumulatively, these results suggest that both PIs act as co-factors to sensitize HPV infection of cervical tissues. Further, differences in progeny virus titers were noted with regards to Kaletra treatment in tissues isolated from different anatomic sites.

### 2.7. Amprenavir Treatment of Primary Cervix Tissue Allows for Virus Transit through Layers for Infecting Basal Cells

Similar to primary gingiva tissues, cervical tissues not treated with PIs were poorly infected with HPV16 compared with tissues that were drug treated (Figure 15A and Figure 16A). In order to visually correlate HPV16 infection with virus localization within different layers of the cervical epithelium, we used *P*-HPV16-BrdU to infect Amprenavir treated primary cervical tissues and performed immunofluorescent staining of virus particles in transit through the tissue over a period of 12–72 h post-infection (Figure 19A,B). At 12 h post-infection, HPV16-BrdU was mostly localized in the cornified and suprabasal layer in tissues not treated with Amprenavir, whereas HPV16-BrdU was localized within the suparabasal layers in drug-treated tissues. At increasing times post-layering of virus on top of tissues, in drug treated tissues, HPV16-BrdU was localized throughout the suprabasal layer as well as within the basal cells. In contrast, virus particles were impeded in the upper cornified layers of untreated tissues. Significant co-localization was determined between viral genome/capsid complexes manifesting as punctate spots in perinuclear regions in basal cells (Pearson’s correlation coefficient for co-localization between 0.578 ± 0.063 to 0.613 ± 0.072) across 12–48 h, followed by a decrease to non-significant levels at 72 h (Pearson’s correlation coefficient for co-localization 0.365 ± 0.0615) post-layering of virus on tissues. These kinetics suggest that at 72 h post-addition of virus, HPV16 in infected basal layer cells of cervical tissues have undergone significant disassembly. This observation is in contrast to gingiva tissues where significant co-localization of viral genome/capsid complexes were observed across all times analyzed including the 72 h time point (Figure 14A,B). These results also suggest that target epithelium from different anatomic sites may regulate kinetics of virus infection of basal cells and downstream establishment of genome replication. Taken together, our results suggest that Amprenavir de-regulation of cell-cell barrier integrity in cervical tissue could play a role in enhancing HPV16 transit through the epithelium to the basal layer for infection.

### 2.8. Progeny HPV16 Can Be Serially Propagated in the Organotypic Epithelium Model

Use of three-dimensional cultures has revolutionized propagation of HPV in the laboratory for conducting detailed studies of the viral life cycle. First, this technique enabled the production of any high-risk HPV type for which the cloned viral genome is available [46,47,48,49]. Viral genomes are electroporated into isolated human mucosal or cutaneous keratinocyte of choice, resulting in chronically infected cell-lines that stably maintain viral episomes. Immortalized cell lines are then differentiated in organotypic cultures for producing infectious HPV stocks. Second, high-risk HPV positive cell lines are generated via acute infection of target epithelial cells that can be grown in three-dimensional cultures for virus production (Chatterjee and Meyers, manuscript in preparation). Third, in the current study, we show that PI treated stratified epithelium can be infected with HPV16 resulting in de novo biosynthesis of infectious progeny virus. Our techniques developed in the current study are important tools to study the mechanistic role of anti-retroviral drugs in promoting HPV infections in HAART-naïve primary epithelium. A related long-term goal is to also understand how changes in viral load may promote new infections of surrounding healthy tissue, thereby potentially creating HPV reservoirs that increase virus persistence, and increase the risk of oral and cervical cancer development in HIV-positive patients undergoing long-term HAART treatment. Therefore, we further asked whether Amprenavir-treated cervical epithelium could be used for serial propagation of virus, such that *prog*-HPV16 isolated from one epithelium is used to infect new epithelium. To account for variation in host genetics, organotypic tissues derived from primary cervical epithelial cells isolated from six different hosts, were treated with Amprenavir and infected with HPV16 as described in the scheme presented here (Figure 20A). Resultant progeny CV stocks were concentrated and titered, followed by infecting a new set of tissues derived from another set of six different hosts, that were further treated with Amprevanir in culture, followed by repeating the process twice more. Cumulatively, the end results depict serial infection of 24 individual host tissues. Moreover, variation in progeny viral titers derived in serial infection/passaging is indicative of differences in host genetics (Figure 20B). Our results show that HPV16 can be propagated in the infected epithelial model, albeit in the presence of PI. In comparison, raft tissues in the first set of infections not treated with Amprenavir displayed very low titers and were unable to be used for serial infection. This is significant in terms of defining one mechanism of long-term virus persistence in HIV infected patients undergoing HAART treatment, since *prog*-HPV16 produced in one area of infected epithelium has the potential to infect neighboring uninfected epithelium, that could eventually translate to creation of HPV reservoirs, virus persistence and cancer progression.

## 3. Discussion

In the current study, we report for the first time that three-dimensional tissues derived from primary epithelial oral and cervical cells can be productively infected with authentic HPV16 in vitro. Our ability to reproduce in vitro human epithelium capable of replicating the complete HPV life cycle provides an opportunity to investigate the effect of HAART/PI as potential co-factors that modulate infection and viral load, factors that determine HPV persistence and cancer of mucosal epithelium. With the advent of HAART, the prevalence of oropharyngeal cancer and persistence of oral warts has increased among HIV patients undergoing anti-retroviral therapy [24]. The underlying cause of increased opportunistic high-risk HPV16 oral infections in patients on HAART treatment is unknown. It is thought that prolonged use of HAART adversely affects turnover rate of the mucosa, which could affect acquisition and establishment of oral disease [24,25,50]. HAART usage is associated with development of adverse oral complications, resulting from oral and perioral manifestations due to oral ulcerations, epithelial hyperplasia and xerostomia [24,25], that damage the mucosal epithelium and potentially expose the underlying tissues to infections due to multiple microorganisms, including HPV infections [24,25,26]. Damage-induced inflammation in the oral epithelium decrease patient adherence to drug regimens [51], that ultimately correlate with suboptimal drug levels and development of drug resistance that could compromise future therapy. Patients on HAART may develop painful oral lesions that affect chewing and swallowing, further contributing to development of malnutrition and weight loss, and also adds to the increased morbidity [52]. The risk of HPV-associated oropharyngeal, cervical and anal cancer are higher among HIV-infected patients in the ART era compared to the general population [16]. The incidence of HPV-associated anal cancer is 80 times, and cervical cancer is 22 times higher in HIV-infected individuals compared to HIV-uninfected individuals [53]. Additionally, HIV-infected individuals have a six-fold greater risk for oropharyngeal and tonsillar cancers than do HIV-uninfected individuals [26]. In contrast, the relationship between the incidence of penile and vulvar cancers and ART exposure have not been reported. However, some studies do indicate HIV infection and immune-compromise as risk-factors for penile cancer, albeit without analyzing whether such patients were undergoing HAART therapy [54]. Another study looked at the relationship between vulvar and other gynecologic cancers in HIV infected women receiving ART, but limiting their analysis to only determining patient survival [55]. Since penile and vulvar cancers also pose significant public health problems in many parts of the developing world, the epidemiology of such neoplasms, in context of availability of HAART therapy, would be expected to vary among different populations. Overall, availability of HAART has extended life of HIV infected patients, but associated with increased incidence of NADC as the leading cause of morbidity and mortality [1,56]. Increasing age/longevity is the greatest risk factor for NADCs, but not sufficient to explain these trends in cancer epidemiology [5].

With ready access to HAART, survival of the population newly infected with HIV is only marginally shorter than that of the HIV-uninfected population [57,58,59]. Consequently, the number of people living with HIV/AIDS has increased four-fold [60,61]. Longer life expectancy afforded by HAART treatment is also associated with increased risk of NADCs compared to the general population [62,63,64,65,66,67]. In 2003, the total number of NADCs exceeded the number of ADCs among people with HIV/AIDS [3]. Cancer deaths were responsible for more than one third of all deaths in HIV-infected patients [17], of which NADCs accounted for 26% of deaths, representing head and neck cancers (8%) and anal cancer (8%), among other malignancies [68]. As a result, NADCs currently comprise the majority of the global burden of cancer in HIV-infected populations, and represent an important public health concern. Our studies have begun to provide a handle on molecular links to epidemiology data that first described the impact of long-term ART on HPV associated oral cancers. How these drugs regulate HPV infection of epithelia, viral load and subsequent risk for cancer progression remains to be determined. In particular, our future studies will reveal novel drivers and pathways related to anatomical site specific impact of PIs on the natural history of HPV in HIV+ patients undergoing HAART treatment.

HIV infection and associated immune suppression is linked with patient susceptibility to opportunistic infections [69]. Immunosuppression may play a role during the early stages of oral HPV carcinogenesis. HPV infections are self-limiting; however, virus persistence is increased in HIV-positive individuals due to immune-dysfunction and reduced HPV clearance, including individuals on long-term ART [70]. The higher prevalence of oral HPV infection among HIV-infected patients could be explained by an increased risk of incident infection due to immunosuppression rather than by reduced clearance [71,72,73]. Importantly, direct effects of HIV-1 transactivator protein tat and gp120 have been shown to modulate disruption of tight junctions in oral mucosal epithelium, that could facilitate HPV infection and reduce clearance [74,75], thereby suggesting a potential mechanism of HPV entry and infection of oral tissue. On the flip-side, one study suggested that HAART treatment could itself abrogate the barrier function of the oral epithelium, thereby increasing invasiveness, and thus malignancy of the HPV-infection [76]. In support of these observations, one study demonstrated that HIV infected patients with an undetectable HIV load had a six-fold risk of presenting HPV oral lesions [11]. Understanding the mechanisms of HPV infection via the paracellular route in HAART treated tissues would provide future opportunities to identify novel/alternate pathways of epithelial cell entry/infection, and key proteins utilized by HPV in this process, as well as delineate mechanisms of HPV persistence in HIV+ patients undergoing therapy. Clearance rates of oral HPV infections in HIV-positive patients are also determined by such factors as sexual behavior and immunosuppression that increase the risk of oral HPV infections [26]. HPV acquisition is increased by high-risk sexual behavior in populations considered at higher risk of acquiring HIV [74]. Alternatively, the high HPV detection rates could be due to increased HPV replication and/or persistence rather than increased HPV acquisition. If persistence of oral HPV leads to HPV-related disease, similar to the genital tract, then increased persistence of HPV could explain the increased prevalence or oral warts in HAART treated HIV+ individuals [15]. Therefore, treatment of HIV, rather than HIV immunosuppression, potentially plays a role in HPV infections in HIV infection [15].

Persistent HPV infections are a major risk to carcinogenic progression. Long intervals between initial infection and the development of cancer imply cofactors are involved. Co-factors that increase infectivity, viral load, and persistence all increase the risk of cancer. We propose that HAART/PI is a novel class of co-factors that regulates HPV infection, and subsequent viral load that could determine persistence and cancer risk. Our ongoing studies focus on mechanisms of HAART induced molecular changes that favor opportunistic HPV infections and changes in HPV load documented in HIV infected patients undergoing treatment (Alam et al., manuscript in preparation). Our studies utilizing the organotypic tissues provide a foundation for understanding mucosal wound healing and regulation of epithelial barrier integrity that promote HPV infection and provide future opportunities to study cellular mechanisms that control HPV infection of epithelial cells. Additionally, our studies also have the potential for future design of novel HIV therapeutics that could protect integrity of the epithelial cell-cell adhesion that minimize opportunistic HPV infections in HIV+ patients. Our in vitro culture system could be applicable as a screening platform for different classes of HAART drugs and their ability to sensitize mucosal epithelia to promote opportunistic HPV infections and subsequent viral load. Identification of infection pathways via the paracellular route could also be used to design therapeutics that minimize the risk of opportunistic HPV infections of target tissues in HIV+ patients undergoing HAART treatment.

## 4. Materials and Methods

### 4.1. Isolation of Gingival and Cervical Keratinocytes

Gingiva tissue was obtained from patients undergoing dental surgery [30]. Cervix tissue was obtained from patients undergoing hysterectomy. To maintain confidentiality, tissue samples were devoid of any identification, such as name, race and age. Approval to collect patient samples as “discarded tissues” was obtained from the Penn State University College of Medicine Institutional Review Board (IRB# 25284). Mixed pools of epithelial cells were isolated from tissues as previously described [30]. Briefly, the connective tissue and dermis were removed from the epithelium and discarded. The epithelial tissue was washed three times with phosphate-buffered saline (PBS) containing 50 µg/mL Gentamycin sulfate (Gibco BRL, Bethesda, MD, USA) and 2× Nystatin (Sigma Chemical Co., St Louis, MO, USA). The epithelial tissue was then minced with scissors and trypsinized into a single-cell suspension using a spinner flask. The suspension was removed, 20 mL of E medium containing 5% fetal calf serum (FCS) was added and cells were pelleted using centrifugation. The supernatant was aspirated and the cell pellet was resuspended in 1 mL of 154 Medium (Cascade Biologics Inc., Portland, OR, USA) supplemented with the Human Keratinocyte Growth Supplement Kit (Cascade Biologics, Inc.) followed by adding to a 10-cm tissue culture plate containing an additional 7 mL of 154 medium. To the spinner flask, 20 mL of fresh trypsin was added to remaining tissue to obtain a second and third round of single-cell suspension. When the cultures became ~70% confluent, they were split 1:3. When cells of the first passage were 70% confluent, the cells were used for growing raft cultures.

### 4.2. Growth of Keratinocytes in Organotypic Cultures

Raft cultures were grown as previously described [30]. Briefly, mouse fibroblast 3T3 J2 were trypsinized and resuspended in 10% reconstitution buffer, 10% 10× DMEM (Dulbecco’s Modified Eagle Medium) (Life Technologies, Gaithersburg, MD, USA), 2.4 μL/mL of 10M NaOH, and 80% collagen (Dickinson, Franklin Lakes, NJ, USA). Cells were added at a concentration of 2.5 × 10^5^ cells/mL. The mixture when then aliquoted into 6 well plates at 2.5 mL per well and incubated at 37 °C for 2–4 h to allow solidification of the collages matrices. Two mL E-media was then added to each to well to allow the matrix to equilibrate. Human gingiva and cervical epithelial cells were trypsinized and resuspended at 2 × 10^6^ cells/mL in E-media and 1 mL of cell suspension was added to each well of the 6 well plate on top of the collagen matrices. Epithelial cells were allowed to attach to the dermal equivalent for 2 h in the presence of 0.005 µg/mL EGF (Epidermal Growth Factor). After removal of the media, the collagen matrices were lifted onto stainless steel grids at the air-liquid interface. The raft cultures were fed by diffusion from below with E-media without EGF for 7 days. On day 8, the rafts were treated with Amprenavir and Kaletra using concentrations and treatment times as described herein. Control raft tissues were fed with E-media and 0.01% ethanol. Raft tissues were fed and treated every other day. Raft tissues were harvested at times as discussed herein.

### 4.3. Protease Inhibitors

Kaletra capsules (200 mg/50 mg) (Lopinavir/ritonavir, Abbott Laboratories) were purchased from the pharmacy at the Milton S Hershey Medical Center, Penn State University College of Medicine. One tablet was crushed into powder and stock solutions were prepared in 70% ethanol. Appropriate dilutions were prepared in E-Media to reach the correct final concentrations prior to feeding the cultures. Amprenavir powder was obtained through the NIH AIDS Reagent Program (Fisher Bioservices, MD, USA) as mentioned in the Acknowledgements sections.

### 4.4. HPV16 Infection of Primary Raft Tissues

Standard laboratory stocks of HPV16 were prepared as described below. Virus stocks were diluted in 200 μL E-Media without serum and gently added drop-wise on top of raft tissues. The beaded droplets were carefully coalesced with a pipette tip to form a uniform layer without disturbing the epithelium.

### 4.5. Transmission Electron Microscopy

Whole raft tissues were harvested without the attached collagen layer and were fixed with 2.5% glutaraldehyde and 2% paraformaldehyde in 0.1 M cacodylate buffer (pH 7.4), and further fixed in 1% osmium tetroxide in 0.1 M cacodylate buffer (pH 7.4) for 1 h. Samples were dehydrated in a graduated ethanol series: pure acetone embedded in LX-112 (Ladd Research, Williston, VT, USA). Thin sections (60 nM) were stained with uranyl acetate and lead citrate and viewed in a JEOL JEM1400 Transmission Electron Microscope (JEOL USA Inc., Peabody, MA, USA).

### 4.6. Production of HPV16 Laboratory Stocks in Organotypic Raft Cultures

Immortalized cervical keratinocytes stably maintaining HPV16 genomes were cultured with J2 3T3 feeder cells and maintained in E-medium and further used for growing the standard 20 day organotypic cultures. Immortalized human cervical keratinocytes persistently infected with HPV16 (cell line HCK16-8) were seeded (1 × 10^6^ cells) onto each collagen matrix consisting of rat-tail type 1 collagen and containing J2 3T3 feeder cells. Following cell attachment and growth to confluence, the matrices were lifted onto stainless steel grids and fed with E-medium supplemented with 10 μM 1,2-dioctanoyl-*sn*-glycerol (C8:O; Sigma Chemical Company, St. Louis, MO, USA) via diffusion from below, as previously described [41]. Raft cultures were allowed to stratify and differentiate for 20 d. Raft cultures were fed every other day, until harvesting tissues on day 20. For BrdU labeling of virions, BrdU (5-Bromo-2′-deoxyuridine) (cat# B5002 Sigma) was added to the media starting on day 8 of raft growth at a final concentration of 25 μM, and replenished during feeding every other day, until harvesting on day 20, as described in this manuscript. Virus stocks were further prepared and titers determined as described below.

### 4.7. HPV16 Isolation and Optiprep Purification of Virions

HPV infected raft tissues were harvested as described [41]. For preparing CV stocks, two rafts were Dounce homogenized in 500 μL of phosphate buffer (0.05 M sodium phosphate [pH 8.0], 2 mM MgCl_2_). Homogenizers were rinsed with an additional 250 μL of phosphate buffer. Non-encapsidated viral genomes were digested by the addition of 1.5 μL (375 U) of benzonase to 750 μL of virus preps, followed by incubation at 37 °C for 1 h. Samples were adjusted to 1 M NaCl by adding 188 μL of ice-cold 5 M NaCl. Samples were further vortexed and centrifuged at 4 °C at 10,500 rpm for 10 min. The supernatants (CV stocks) were stored at −80 °C for further analysis. Optiprep purification: Optiprep purification of CV stocks was performed as previously described [41]. Briefly, Optiprep gradients were prepared by underlaying 27%, 23% and 39% Optiprep. Gradients were allowed to diffuse for 1 h at room temperature. Then 300 μL of clarified, benzonase-treated CV stock was layered on top of the gradient. Tubes were then centrifuged in a SW55 rotor (Beckman, Pasadena, CA, USA) at 234,000× *g* for 3.5 h at 18 °C. After centrifugation, 11–500 μL fractions were carefully collected, top to bottom, from each tube. Virus titers in fractions were determined as described below. Where specified, CV stocks were concentrated using Amicon^®^ Ultra-4 Centrifugal Filters (30 K) (Merck Millipore, Burlington, MA, USA). Samples were centrifuged for 30 min at 3000 rpm, and stored at −80 °C for further analysis.

### 4.8. Titering HPV16 Virus Stocks

HPV16 titers were measured using qPCR-based DNA encapsidation assay as previously described [41]. To detect endonuclease-resistant genomes in CV stocks and Optiprep fractions the following method was used. Briefly, viral genomes were released from 10 μL benzonase-treated CV stock or 20 μL Optiprep fraction by re-suspension in 200 μL HIRT DNA extraction buffer (400 mM NaCl/10 mM Tris-HCl (pH 7.4)/10 mM EDTA (pH 8.0)), 2 μL 20 mg/mL proteinase K, and 10 μL 10% SDS, for 2–4 h at 37 °C. Following digestion, the DNA was extracted twice using phenol-choloroform-isoamyl alcohol (25:24:1), followed by extraction in an equal amount of chloroform. DNA was ethanol precipitated overnight at −20 °C. Samples were centrifuged, and the DNA pellet was washed with 70% ethanol and resuspended in 20 μL of Tris-EDTA overnight. To quantify viral genomes, a Thermo Scientific Maxima SYBR Green qPCR kit was utilized. Amplification of the HPV16 E2 open reading frame (ORF) was performed using 0.3 μM of forward primer HPV16E2-5′ and HPV16E2-3′ (Appendix A). Amplification of the E2 ORF of serially diluted pBSHPV16 DNA, ranging from 10^8^–10^4^ copies/μL, was used to generate a standard curve. A Bio-Rad iQ5 Multicolor Real-Time qPCR machine and software were utilized for PCR amplifications and subsequent data analysis.

### 4.9. RT-qPCR Infectivity Assays in HaCaT Monolayer Cultures

All infectivity studies were performed using HaCaT keratinocytes. HaCaT cells were seeded 50,000 cells/well in 24-well plates and infectivity assays were performed as previously described [41]. Briefly, cells were incubated with virus (CV stocks using MOI of 10, or Optiprep fraction #7 using MOI of 1) samples in cell culture medium for 48 h at 37 °C in 5% CO2 followed by mRNA harvesting using the RNeasy kit (Qiagen, Hilden, Germany). Infections were analyzed using a RT-qPCR based assay detecting levels of the E1^E4 splice transcript (QuantiTect Probe RT-PCR Kit). The HPV16 E1^E4 transcript was detected using 4 μM of the forward primer HPV16E1^E4-5′ and reverse primer HPV16E1^E4-3′ and using 0.2 μM of HPV16 E1^E4 fluorogenic probe (Appendix A). The TATA-binding protein (TBP) amplicons were detected using 0.125 μM primers TBP-5′ and TBP-3′, and 0.2 μM of fluorogenic probe (Appendix A). For each sample, the E1^E4 transcript abundance was normalized to TBP using infection of standard HPV16 laboratory stocks (either CV stock or Optiprep fraction #7) as controls, arbitrary designated as 1, using MOIs as described herein.

### 4.10. Virus Neutralization and Infectivity Assays in HaCaT Monolayer Cultures

For neutralization assays, virus samples were co-incubated with antibodies in 500 μL culture medium for 1 h at 37 °C prior to infecting HaCat monolayer cultures followed by RNA harvesting as described above. For these experiments, conformation-dependent anti-L1 mouse antibody H16.V5 (1:1000 dilution; a kind gift from Neil Christensen, Penn State College of Medicine) or the anti-L2 mouse antibody RG-1 (1:500 dilution; a kind gift from Richard Roden, John Hopkins) were used. RNA samples were harvested followed by the infectivity assay as described above [41].

### 4.11. RNA Isolation from Raft Tissue Samples and RT-qPCR to Determine E1^E4 Transcript Expression in Infectivity Assays

DNase I treated total RNA was isolated from 30 mg raft tissue using the RNeasy Fibrous Tissue Mini Kit (Qiagen cat. No. 74704), as per instructions provided by the manufacturer. Infections were analyzed using RT-qPCR based assay detecting levels of the E1^E4 splice transcript and TBP amplicons as described above. Untreated tissues infected with the highest dose of HPV16 virions were used as controls. Expression abundance of the E1^E4 spliced transcript in HPV16 infected/drug treated tissues was normalized to TBP and relative levels compared to HPV16 infected/non-drug treated tissues as controls (arbitrary designated as 1).

### 4.12. Immunofluorescence Analysis of BrdU-Labeled HPV16 Virions Infecting HaCat Monolayer Cultures

HaCaT monolayer cultures were plated on glass coverslips and incubated 10–12 h when cells reached 70% confluency. Cells were infected with HPV16-BrdU or *prog*-HPV16-BrdU Optiprep gradient [F#7] stocks using an MOI of 1. Infected cells were incubated for times as described herein. Post-infection, cells were fixed in 4% *w/v* phosphate buffered-paraformaldehyde (pH 7.4) for 10 min, washed three times in PBS, and permeabilized with 0.2% Triton X-100 in PBS for 10 min, followed by washing three times in PBS. Coverslips were blocked with 5% goat serum/1% bovine serum albumin in PBS for 20 min followed by co-incubation with anti-BrdU anti-mouse IgG1 (IIB5, Abcam) (1:1000) and anti-L1 anti-rabbit antibody (1:1000) (a kind gift from Dr. Neil Christensen, Penn State College of Medicine) for 60 min. Coverslips were washed three times with PBS and further co-incubated with secondary antibodies Alexa Fluor 488-conjugated anti-mouse IgG (2 µg/mL; Invitrogen, Carlsbad, CA, USA) (1:1000) and Alexa Fluor 568-conjugated anti-rabbit IgG (1 µg/mL; Invitrogen) (1:1000) in the absence of serum. Coverslips were washed three times with PBS and nuclei were stained with Hoechst (1:5000) for 10 min. Cells were washed twice with PBS, and mounted in ProLong Diamond (Invitrogen). Images were taken on a C2+ confocal microscope system (Nikon, Melville, NY, USA). Images were processed using NIS Elements software. Images also show volume renderings of z-stacks. Pearson’s coefficients were determined and statistical analysis was performed on 5 separate images.

### 4.13. Immunofluorescence Analysis of Raft Tissue Sections Post-Layering with HPV16-BrdU

Raft cultures were treated with Amprenavir and followed by layering of HPV16-BrdU on top of tissues for 12, 24, 48 and 72 h, as indicated followed by harvesting. Tissues were fixed in 10% buffered formalin, embedded in paraffin and 4 μM sections were prepared. For immunofluorescence staining, the slides were submerged in xylene for de-paraffinization and then rehydrated. Antigen retrieval was achieved by submerging the slides in Tris-EDTA buffer (pH 9) in a 90 °C water bath for 10 min. The slides were then rinsed with Tris-buffered saline (TBS)-Tween and blocked with Background Sniper blocking reagent (Biocare Medical, Pacheco, CA, USA). The slides were then stained with the primary antibodies (anti-BrdU and anti-HPV16 L1, 1:1000 dilution each antibody) overnight at 4 °C. The slides were then rinsed with TBS-Tween 3 times, 10 min each, and stained with secondary antibody (Alexa Fluor 488 and Alexa Fluor 568; (Life Technologies, Carlsbad, CA, USA) diluted 1:1000 for 1 h at room temperature. The slides were then rinsed once with (TBS)-Tween, followed by staining with Hoechst nuclear stain (1:5000) dilution for 5 min, and then rinsed twice with TBS-Tween. All primary and secondary antibodies were diluted in Da Vinci Green diluent (Biocare Medical). Slides were washed with PBS, and mounted in ProLong Diamond (Invitrogen). Images were taken on a C2+ confocal microscope system (Nikon). Images were processed using NIS Elements software (Nikon, Melville, NY, USA).

### 4.14. Statistical Analysis

Data were analyzed using Prism 8.0 by Graphpad (La Jolla, CA, USA). Quantitative data are presented as mean ± standard deviation. Significance was based on pairwise Student’s *t*-test. Comparisons with *p* > 0.05 are indicated by NS (not significant); 0.01 < *p* < 0.05 by *; 0.001 < *p* < 0.01 by **; 0.0001 < *p* < 0.001 by ***; and *p* < 0.0001 by ****.

## 5. Conclusions

The risk of HPV-associated oropharyngeal, cervical and anal cancers are higher among HIV-infected patients in the ART era compared to the general population [16]. Generally, HPV infections are self-limiting; however, persistent HPV infection is a major risk to carcinogenic progression. Prolonged use of HAART adversely affects turnover rate of the oral epithelium, leading to oral complications that could affect acquisition and establishment of HPV infection and oral disease, and the same mechanism could affect turnover of the cervical epithelium. Our results presented in this manuscript indicate that HAART treatment creates favorable cellular conditions for opportunistic HPV infections in target epithelium. The organotypic raft tissues can physiologically model carcinogenic stages from precancerous to cancer [77,78,79]. Using this system, our ability to reproduce in vitro human epithelium capable of replicating the complete HPV life cycle is an opportunity to investigate how HAART manipulates normal cellular mechanisms and signaling pathways to promote HPV16 infection and de novo virus biosynthesis, which is an important milestone in driving virus persistence. We propose that HAART is a potential co-factor that modulates HPV infection and subsequent changes in viral load that could determine viral persistence and cancer of the oral cavity and cervix, and potentially the anal canal. Our future studies are geared towards mapping the molecular interaction of HAART with the drug-naïve primary epithelium, and how this interaction affects downstream cellular targets that regulate HPV infection, subsequent viral load and cancer progression.

## Figures and Tables

**Figure 1 cancers-12-02664-f001:**
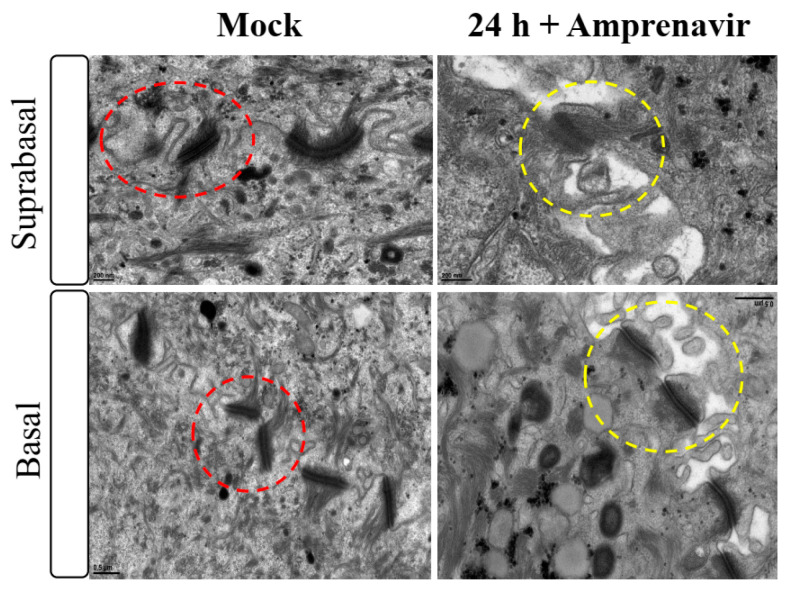
Transmission electron microscopy depicting cell-cell junction morphology of day 8 gingiva raft tissue treated with Amprenavir (7.66 µg/mL) for 24 h compared with tissue not drug treated.

**Figure 2 cancers-12-02664-f002:**
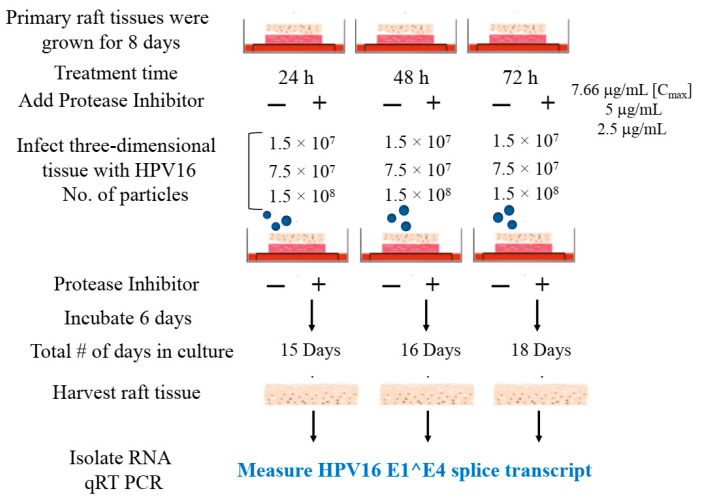
Schematic of raft tissue growth, drug treatment and infection using three doses of standard laboratory stocks of HPV16 virions.

**Figure 3 cancers-12-02664-f003:**
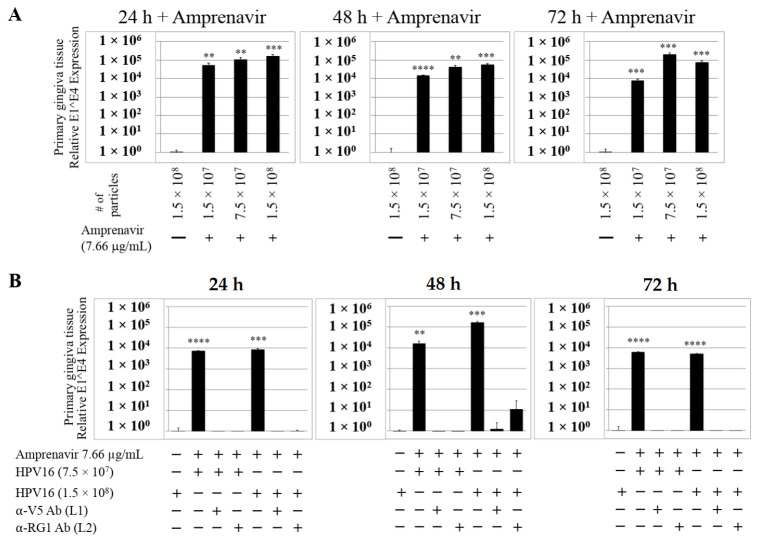
Amprenavir (7.66 µg/mL) treatment sensitizes primary gingiva tissue to HPV16 infection (**A**) Comparative expression of HPV16 E1^E4 transcripts in drug treated tissues compared with virus infected tissues not drug treated. (**B**) Inhibition of virus infection of tissues using HPV16 pre-incubated with α-V5 and α-RG1. Data were analyzed as mean ± SD. *p*-values were calculated using two-tailed Student’s *t*-tests. Quantitative data are presented as mean ± standard deviation. Significance was based on pairwise Student’s *t*-test. Comparisons are indicated as 0.001 < *p* < 0.01 by **; 0.0001 < *p* < 0.001 by ***; and *p* < 0.0001 by ****.

**Figure 4 cancers-12-02664-f004:**
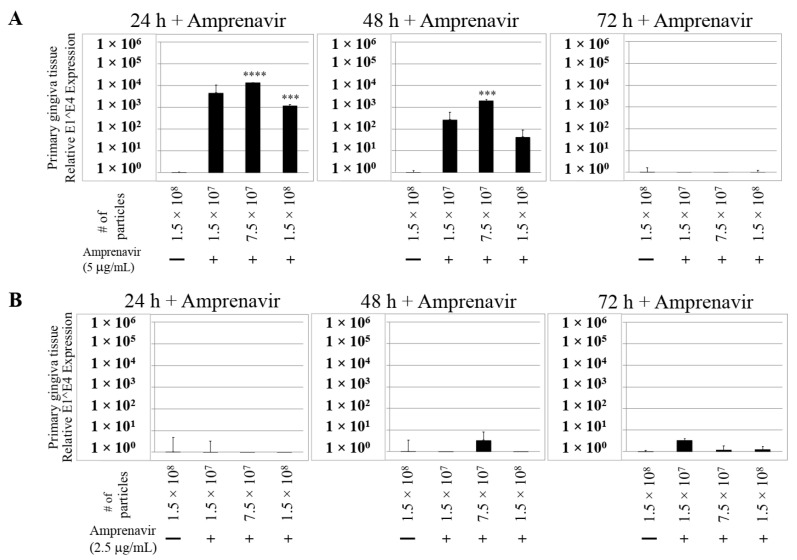
Dose-dependent Amprenavir treatment modulates HPV16 infection of gingiva tissues. (**A**) Comparative expression of HPV16 E1^E4 transcripts in tissues treated with 5 µg/mL Amprenavir. Results shown are average of three individual experiments. (**B**) Comparative expression of HPV16 E1^E4 transcripts in tissues treated with 2.5 µg/mL Amprenavir. Data were analyzed and is presented as mean ± SD. *p*-values were calculated using two-tailed Student’s *t*-tests. Quantitative data are presented as mean ± standard deviation. Significance was based on pairwise Student’s *t*-test. Comparisons are indicated as 0.0001 < *p* < 0.001 by ***; and *p* < 0.0001 by ****.

**Figure 5 cancers-12-02664-f005:**
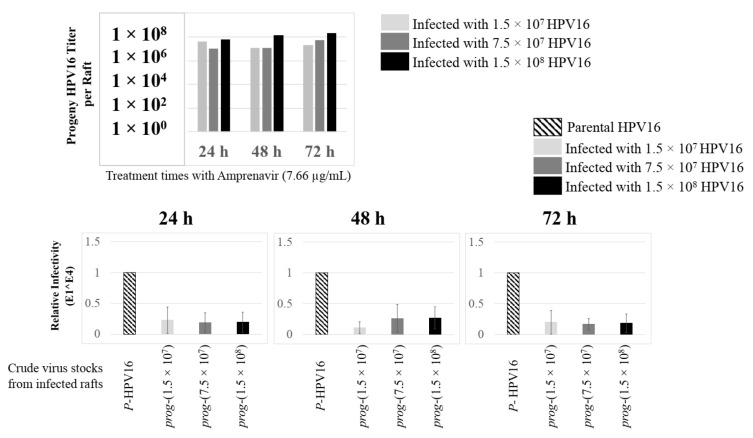
Progeny HPV16 stocks isolated from day 18 gingiva tissues treated with Amprenavir are poorly infectious. Top panel: Progeny HPV16 virus stock (day18 harvest) titers isolated from raft tissues infected with three virus doses indicated in light grey bars: 1.5 × 10^7^ HPV16 virions; Grey bars: 7.5 × 10^7^ HPV16 virions; Black bars: 1.5 × 10^8^ HPV16 virions. Bottom panel: Corresponding infectivity of progeny virion stocks (5 MOI) (Multiplicity of Infection) in HaCaT monolayer cells compared with standard (Parental) laboratory stocks. Infection results shown are average of three experiments and is presented as mean ± SD.

**Figure 6 cancers-12-02664-f006:**
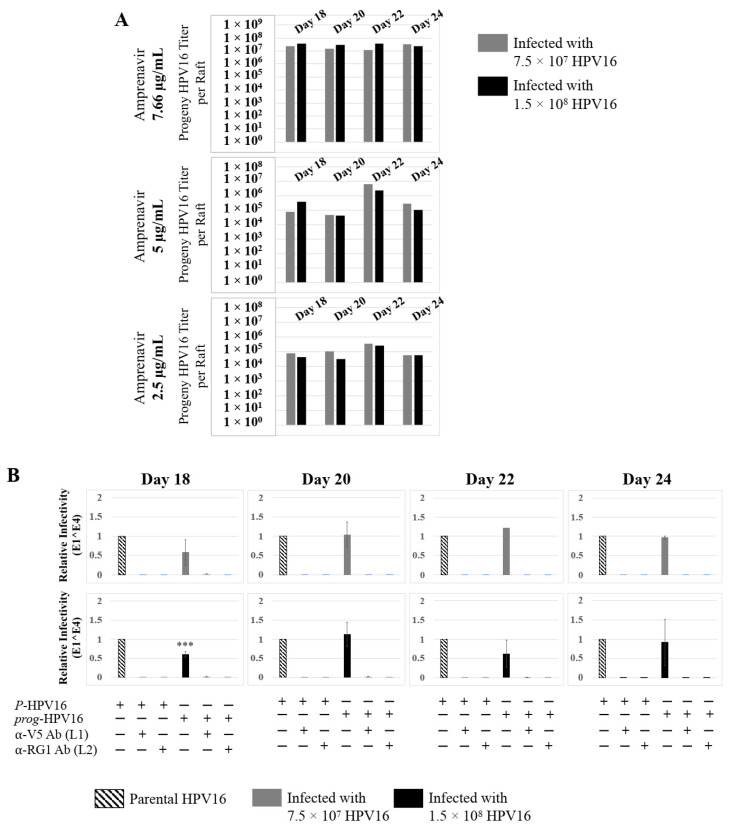
Extended culturing of infected gingiva tissues treated with Amprenavir determines progeny virus titers. (**A**) Raft tissues (day 18–24) infected with two virus doses modulate *prog*-HPV16 titers in an Amprenavir concentration dependent manner. Grey bars: infected with 7.5 × 10^7^
*P*-HPV16 virions; Black bars: infected with 1.5 × 10^8^
*P*-HPV16 virions. (**B**) Infectivity of *prog*-HPV16 Optiprep [F#7] compared with *P*-HPV16 [F#7] (1 MOI), and infection inhibition using α-V5 and α-RG1 monoclonal antibodies. Infection results shown are average of three experiments and is presented as mean ± SD. *p*-values were calculated using two-tailed Student’s *t*-tests. Infectivity of *prog*-HPV16 were not significantly different compared with *P*-HPV16. Comparisons are indicated as 0.0001 < *p* < 0.001 by ***.

**Figure 7 cancers-12-02664-f007:**
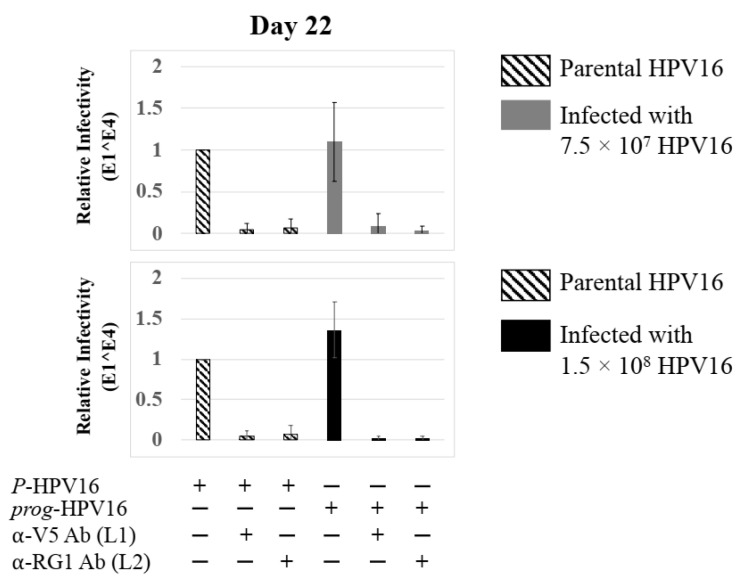
Low Amprenavir concentrations determine production and infectivity of *prog*-HPV16. Optiprep [F#7] *prog*-HPV16 from tissues treated with 5 µg/mL Amprenavir compared with *P*-HPV16 Optiprep [F#7] (1 MOI in HaCaT cells), and infection inhibition using α-V5 and α-RG1 monoclonal antibodies. Infection results shown are average of three experiments and is presented as mean ± SD. *p*-values were calculated using two-tailed Student’s *t*-tests. Infectivity of *prog*-HPV16 was not significantly different compared with *P*-HPV16.

**Figure 8 cancers-12-02664-f008:**
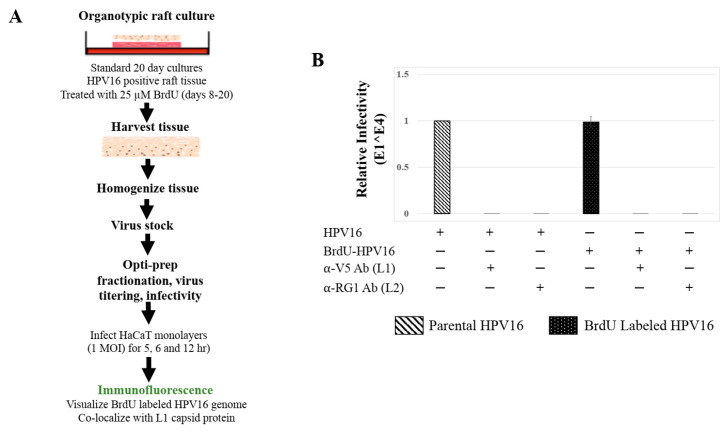
BrdU Labeling of 20 d Standard Laboratory HPV16 Stocks. (**A**) Schematic for BrdU labeling of HPV16 virus in 20 d raft tissues derived from human cervical cell line, maintaining episomal HVP16 genomes. (**B**) Infectivity comparison of BrdU-labeled HPV16 versus control unlabeled virus stocks (1 MOI in HaCaT cells). Optiprep gradient purified virus [F#7] stocks was used. Infection was in the presence of α-V5 and α-RG1 monoclonal antibodies. Infection results shown are average of three experiments and is presented as mean ± SD. *p*-values were calculated using two-tailed Student’s *t*-tests. Infectivity comparisons were not significantly different.

**Figure 9 cancers-12-02664-f009:**
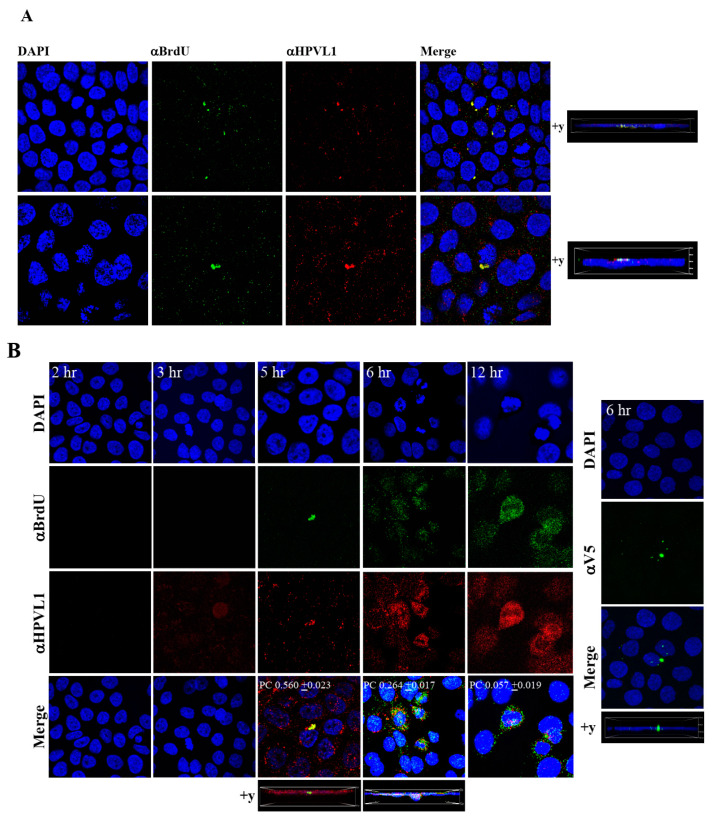
Visualization of BrdU-labeled viral genomes. (**A**) Confocal microscopy imaging of Optiprep [F#7] BrdU-labeled HPV16 genomes co-localized with L1 capsid proteins in HaCaT cells 5 h post-infection. Cells were infected using 1 MOI of virus. (**B**) Left panel: Time-dependent co-localization of HPV16-BrdU labeled genomes and L1 capsids in HaCaT monolayers, 2 h, 3 h, 5 h and 6 h post-infection. Optiprep gradient virus in [F#7] stocks (1 MOI) was used. Right panel: Immunofluorescence of α-V5/L1 capsid protein complexes that correlate with inhibition of infection. Optiprep gradient fractionated [F#7] virus stock was incubated with α-V5/L1 followed by infecting HaCaT monolayers (1 MOI) and imaging 6 h post-infection (compare with left panel, 6h post-infection L1 staining pattern without α-V5 incubation). Pearson’s coefficients illustrating co-localization of BrdU labeled genome and L1 capsid protein are presented in the merged images. Data represent mean Pearson’s coefficient ± SD, calculated from 5 independent images.

**Figure 10 cancers-12-02664-f010:**
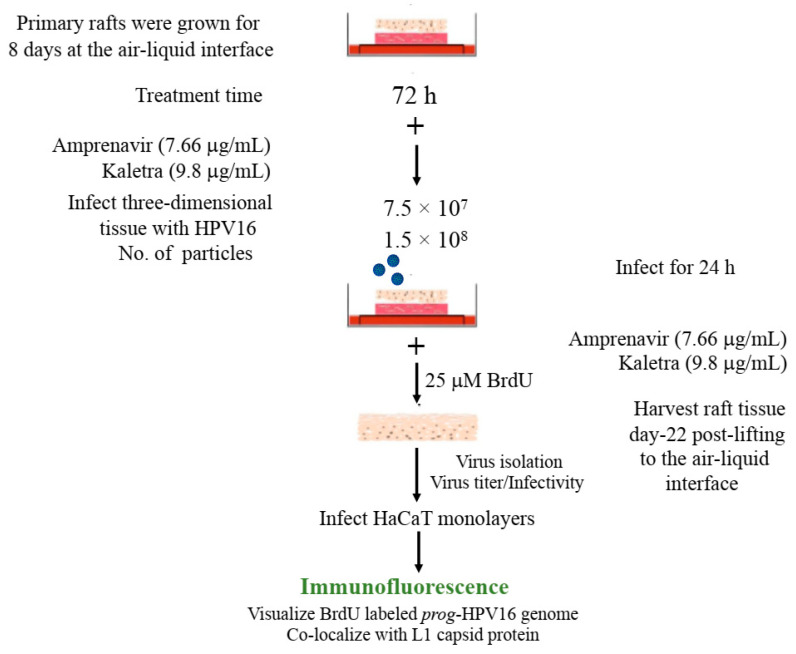
BrdU labeling of progeny HPV16 biosynthesized in primary tissues. Schematic for BrdU labeling of *prog*-HPV16 biosynthesized in 22 d PI treated tissues. In all cases, Optiprep gradient purified [F#7] day 22 virions were used for infections (1 MOI).

**Figure 11 cancers-12-02664-f011:**
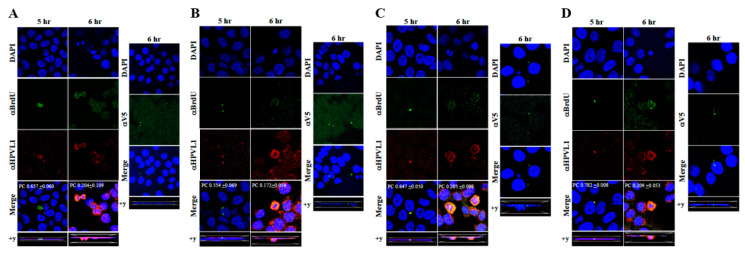
Immunofluorescence co-localization of BrdU-labeled genomes and L1 capsids of *prog*-HPV16. (**A**) Progeny HPV16 biosynthesized in Amprenavir treated gingiva tissues. (**B**) *Prog*-HPV16 biosynthesized in Kaletra treated gingiva tissues. (**C**) *Prog*-HPV16 biosynthesized in Amprenavir treated cervical tissues. (**D**) *Prog*-HPV16 biosynthesized in Kaletra treated cervical tissues. Right panels of (**A**–**D**): Immunofluorescence of α-V5/L1 capsid protein complexes that correlate with inhibition of infection. *Prog*-HPV16 [F#7] stocks biosynthesized in 22 d raft tissues was incubated with α-V5/L1 followed by infecting HaCaT monolayers and then imaging 6 h post-infection (compare with 6 h post-infection L1 staining pattern without α-V5 incubation, left panel). Pearson’s coefficients illustrating co-localization of BrdU labeled genome and L1 capsid protein are presented in the merged images. Data represent mean Pearson’s coefficient ± SD, calculated from 5 independent images.

**Figure 12 cancers-12-02664-f012:**
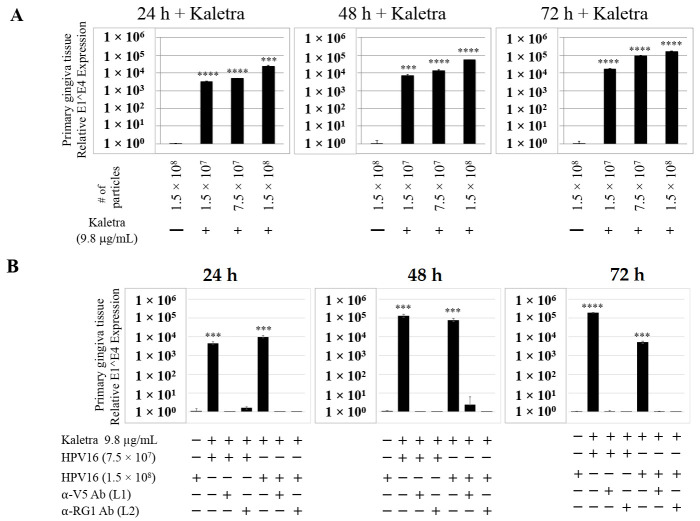
Kaletra (9.8 µg/mL) treatment sensitizes primary gingiva tissue to HPV16 infection. (**A**) Comparative expression of HPV16 E1^E4 transcripts in Kaletra treated tissues compared with virus infected tissues not drug treated. Three *P*-HPV16 doses were used for infecting rafts as indicated. (**B**) Inhibition of virus infection of tissues using HPV16 pre-incubated with α-V5 and α-RG1. Data were analyzed as mean ± SD. *p*-values were calculated using two-tailed Student’s *t*-tests. Significance was based on pairwise Student’s *t*-test. Comparisons are indicated as 0.0001 < *p* < 0.001 by ***; and *p* < 0.0001 by ****.

**Figure 13 cancers-12-02664-f013:**
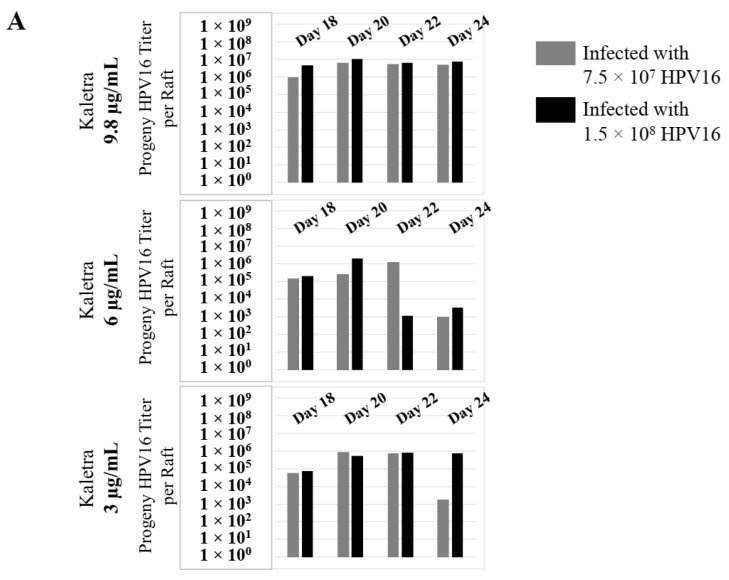
Extended culturing of Kaletra infected gingiva tissues determines progeny virus titers. (**A**) Raft tissues (day 18–24) infected with two virus doses modulates *prog*-HPV16 titers in a Kaletra concentration dependent manner. Grey bars: infected with 7.5 × 10^7^
*P*-HPV16 virions; Black bars: infected with 1.5 × 10^8^
*P*-HPV16 virions. (**B**) Infectivity of *prog*-HPV16 Optiprep [F#7] compared with *P*-HPV16 [F#7] (1 MOI), and infection inhibition using α-V5 and α-RG1 monoclonal antibodies. Infection results shown are average of three experiments and is presented as mean ± SD. *p*-values were calculated using two-tailed Student’s *t*-tests. Infectivity of *prog*-HPV16 were not significantly different compared with *P*-HPV16.

**Figure 14 cancers-12-02664-f014:**
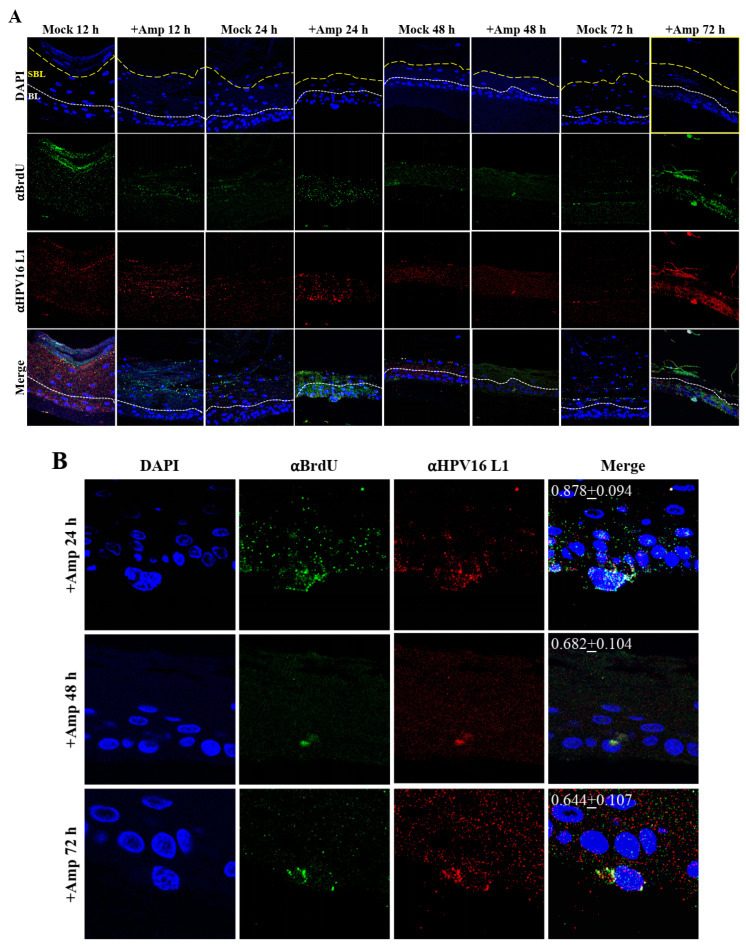
Time-course visualization of HPV16-BrdU transit through primary gingiva tissues. (**A**) Amprenavir (7.66 µg/mL) treated gingiva tissues (72 h) were infected with 5 × 10^6^ HPV16-BrdU virions and tissues harvested and fixed periodically 12–72 h post-layering of virus on top of tissue. Immunofluorescence staining/confocal analysis of tissue sections staining the HPV16-BrdU genomes/L1 capsid complexes within cornified, suprabasal and basal layers. (**B**) 20× magnification of infected basal cells shown in (**A**). Pearson’s coefficients illustrating co-localization of BrdU labeled genome and L1 capsid protein in basal layer cells are presented in the merged images. Data represent mean Pearson’s coefficient ± SD, calculated from 3 independent images.

**Figure 15 cancers-12-02664-f015:**
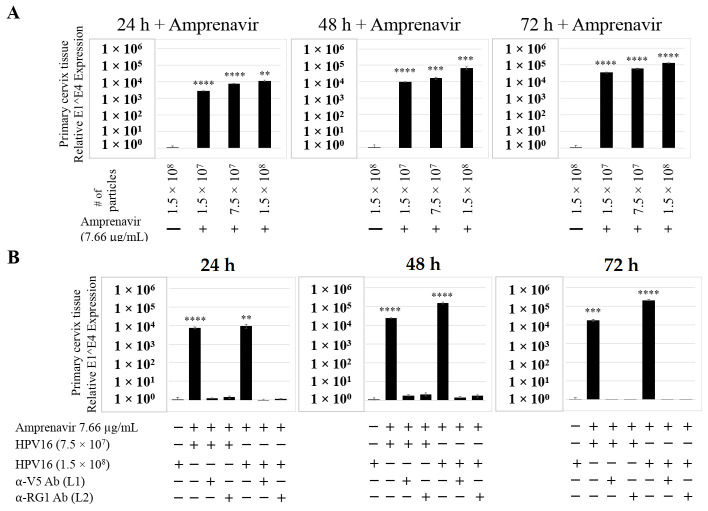
Amprenavir (7.66 µg/mL) treatment sensitizes primary cervical tissue to HPV16 infection. (**A**) Comparative expression of HPV16 E1^E4 transcripts in drug treated tissues compared with virus infected tissues not drug treated. (**B**) Inhibition of virus infection of tissues using HPV16 pre-incubated with α-V5 and α-RG1. Data were analyzed as mean ± SD. *p*-values were calculated using two-tailed Student’s *t*-tests. Comparisons are indicated as 0.001 < *p* < 0.01 by **; 0.0001 < *p* < 0.001 by ***; and *p* < 0.0001 by ****.

**Figure 16 cancers-12-02664-f016:**
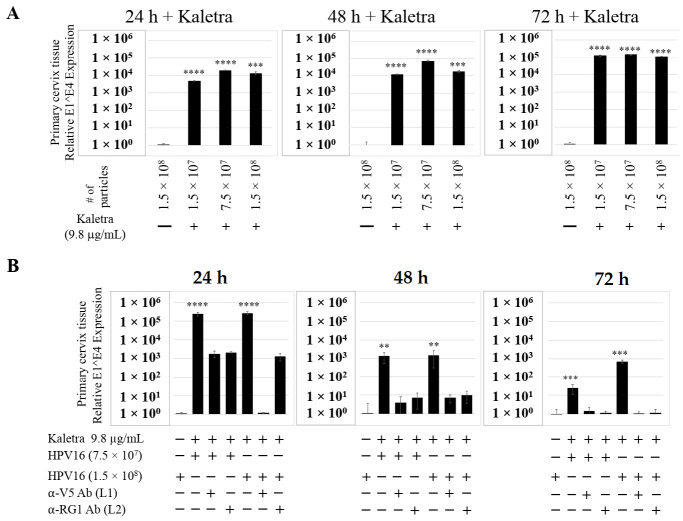
Kaletra (9.8 µg/mL) treatment sensitizes primary cervical tissue to HPV16 infection. (**A**) Comparative expression of HPV16 E1^E4 transcripts in Kaletra (9.8 µg/mL) treated tissues compared with virus infected tissues not drug treated. (**B**) Inhibition of HPV16 infection using α-V5 and α-RG1 of tissues treated with Kaletra. Data were analyzed as mean ± SD. *p*-values were calculated using two-tailed Student’s *t*-tests. Comparisons are indicated as 0.001 < *p* < 0.01 by **; 0.0001 < *p* < 0.001 by ***; and *p* < 0.0001 by ****.

**Figure 17 cancers-12-02664-f017:**
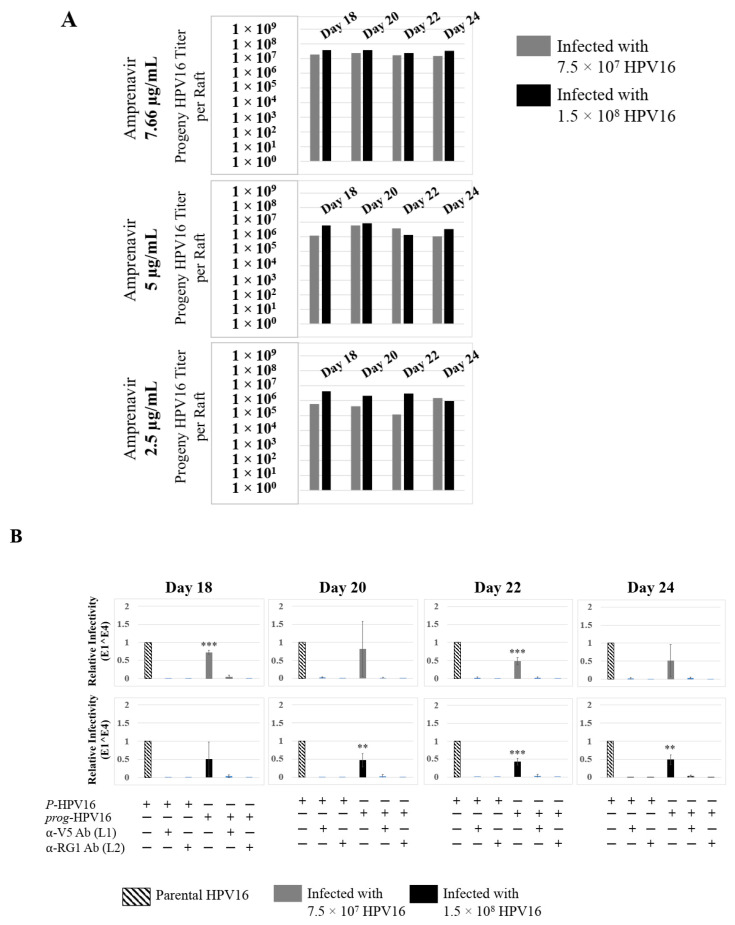
Extended culturing of Amprenavir treated HPV16 infected cervical tissues modulates progeny virus titers. (**A**) Raft tissues (day 18–24) infected with two virus doses modulate *prog*-HPV16 titers in an Amprenavir concentration dependent manner. Grey bars: infected with 7.5 × 10^7^
*P*-HPV16 virions; Black bars: infected with 1.5 × 10^8^
*P*-HPV16 virions. (**B**) Infectivity of *prog*-HPV16 Optiprep [F#7] compared with *P*-HPV16 [F#7] (1 MOI in HaCaT cells), and infection inhibition using α-V5 and α-RG1 monoclonal antibodies. Infection results shown are average of three experiments and is presented as mean ± SD. *p*-values were calculated using two-tailed Student’s *t*-tests. Infectivity of *prog*-HPV16 were not significantly different compared with *P*-HPV16. Comparisons are indicated as 0.001 < *p* < 0.01 by **; 0.0001 < *p* < 0.001 by ***.

**Figure 18 cancers-12-02664-f018:**
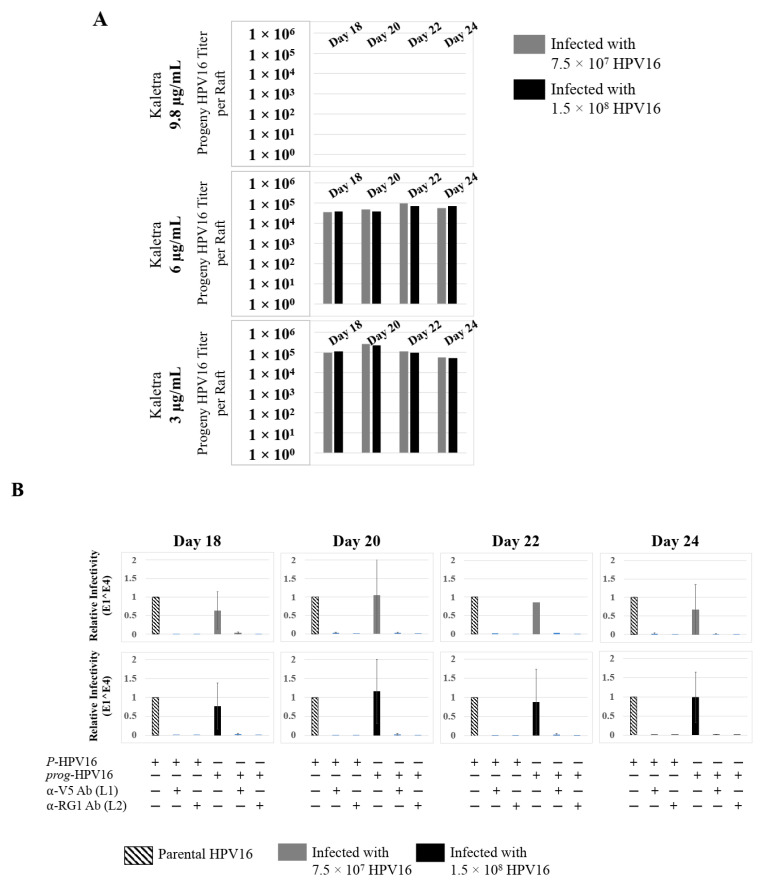
Extended culturing of Kaletra treated HPV16 infected cervical tissues modulates progeny virus titers. (**A**) Raft tissues (day 18–24) infected with two virus doses modulates *prog*-HPV16 titers in a Kaletra concentration dependent manner. Grey bars: infected with 7.5 × 10^7^
*P*-HPV16 virions; Black bars: infected with 1.5 × 10^8^
*P*-HPV16 virions. (**B**) Infectivity of concentrated virus stocks isolated from raft tissues treated with Kaletra (3 µg/mL) compared with *P*-HPV16 (1 MOI in HaCaT cells), and infection inhibition using α-V5 and α-RG1 monoclonal antibodies. Infection results shown are average of three experiments and is presented as mean ± SD. *p*-values were calculated using two-tailed Student’s *t*-tests. Infectivity of *prog*-HPV16 were not significantly different compared with *P*-HPV16.

**Figure 19 cancers-12-02664-f019:**
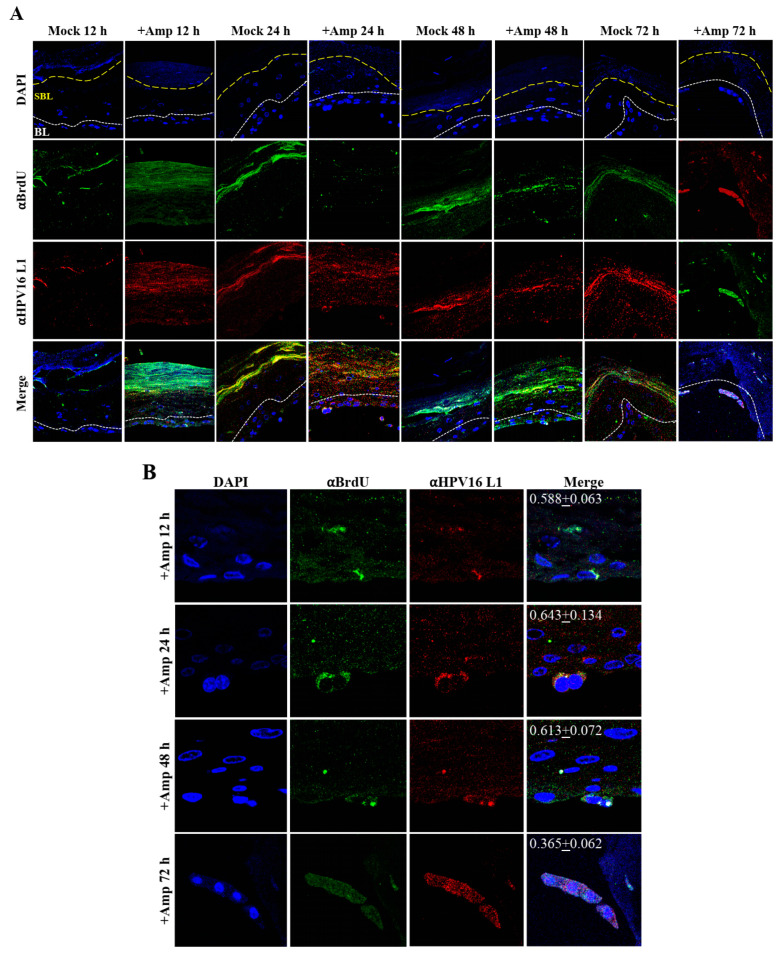
Time-course visualization of HPV16-BrdU transit through primary cervix. (**A**) Amprenavir (7.66 µg/mL) treated cervical tissues (72 h) were infected with 5 × 10^6^ HPV16-BrdU virions and tissues harvested and fixed periodically 12–72 h post-layering of virus on top of tissue. Immunofluorescence staining/confocal analysis of tissue sections staining the HPV16-BrdU genomes/L1 capsid complexes within cornified, suprabasal and basal layers. (**B**) 20× magnification of infected basal cells in (**A**). Pearson’s coefficients illustrating co-localization of BrdU labeled genome and L1 capsid protein in basal layer cells are presented in the merged images. Data represent mean Pearson’s coefficient ± SD, calculated from 3 independent images.

**Figure 20 cancers-12-02664-f020:**
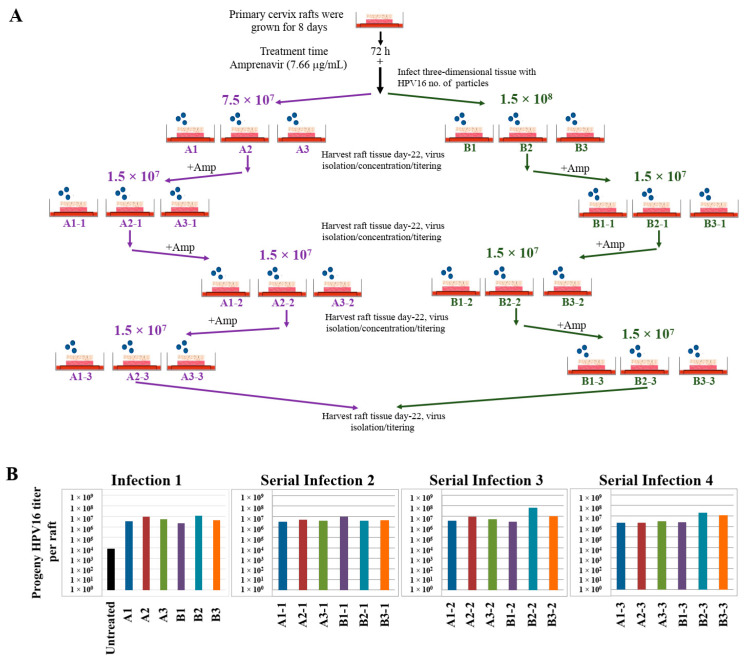
Serial propagation of progeny HPV16 in primary cervical tissues treated with Amprenavir. (**A**) Schematic for serial propagation of HPV16 in Amprenavir (7.66 µg/mL) treated primary cervical tissues. (**B**) *Prog*-HPV16 titers in serial infections representing 24 different host lines. Each infected raft tissue designated in (**A**) was titered and presented in graphs. Each bar is color coded to represent serial infections (Serial infections 3–4) from the same initial infection (Infection 1). Virus stocks were concentrated prior to infecting the next set of raft tissues.

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
