# Peer review of "Anti-Retroviral Protease Inhibitors Regulate Human Papillomavirus 16 Infection of Primary Oral and Cervical Epithelium"

_cancers, 2020, doi:10.3390/cancers12092664_

Round 1
Reviewer 1 Report
The manuscript has been accurately revised by the authors according to my comments and it can be accepted for publication as it is.
Reviewer 2 Report
I am happy that the authors have addressed my concerns.
This manuscript is a resubmission of an earlier submission. The following is a list of the peer review reports and author responses from that submission.
Round 1
Reviewer 1 Report
Review of Cancers-878798
In this manuscript, the authors explore the hypothesis that the increased incidence of certain HPV-related malignancies (HPV+ oropharyngeal squamous cell carcinoma and cervical cancer) observed in HIV+ patients treated with high-activity antiretroviral therapy (HAART) is directly related to an activity of protease inhibition on infectivity. They start by showing evidence that an early protease inhibitor (PI) Aprenavir (no longer used in HAART) induces morphological changes in rafts suggestive of HPV infection in primary epithelial raft cultures (keratinocytes from human gingiva) as measured by HPV E1^E4 expression. The E1^E4 transcript is specific to later stages of HPV infection and is involved in HPV genome amplification in the mid-layers of the epithelium and possibly also in viral packaging and release. Importantly, the authors complement the E1^E4 analysis with measures of progeny viral titre generated from rafts -/+ PI treatment and use BrdU labelling and L1 immunofluorescence to visualise HPV. This is important as it provides a more direct measure of the virus and helps to support the author’s assetion that PI treatment enhances access to the basal layers of the epithelium; the site of infection. The work with Aprenavir is supported by similar observations with another PI (Kaletra). Overall, a number of innovative approaches are taken here and as the authors point out, this is the first demonstration of an organotypic system for infection of primary gingival or cervical keratinocytes with HPV16. This is a useful advance and taken together with the insight into how HAART may increase HPV-related malignancies, this work will be of interest to those studying HPV biology and pathogenesis. I do however, have some concerns regarding a number of the figures.
Specific comments
At the resolution of electron microscopy images provided, I was unable to see clear differences in cell-cell junctions between mock and 24 hour treated rafts, although it does appear there are more ‘gaps’ in the treated rafts. Suggest providing / showing higher resolution micrographs if possible.
The resolution on the graphs thoughout is also poor but from what I could make out, some strange y-axis scales are used that are neither log nor linear (e.g. Fig 1C, Fig 1G). This serves to greatly increase the apparent effects of HPV dose which in reality are small in comparison to the overall effect of -/+ Amprenavir. Similar use of scales occurs throughout the paper.
The resolution of the supplementary figures is even worse and I can’t read the y-axis scales on these figures at all, making it impossible to comment on them. I can appreciate that when working with different primary cultures there will be differences due to genetic background and possibly other factors but from what I can see, at least it should be possible to perform some kind of statistical significance testing across the three independent replicates for a number of these experiments. The differences -/+ Aprenavir look very substantial but again it is difficult to tell due to the illegibility of the scales on the supplementary figures in particular.
The data shown in Figure 5 are important, as they are used to support the hypothesis that PI treatment allows HPV16 to access and infect cells in the basal layer of the epithelium. However, in its current format it is hard to tell from the images provided whether the PI treatment is indeed resulting more HPV16 reaching the basal layer. No quantification is provided and I don’t find the images shown particularly convincing. There is also an error in Figure 5C, in which it appears an anti-L1 image is switched with an anti-BrdU image. The merged images are hard to interpret and care should be taken to ensure all exposures/contrast are kept constant (e.g. in Figure 5A, comparing Mock 48 h with +Amp 48 h, in the merged images the green channel looks much stronger in the +Amp image compared with the minor difference in intensity in the panels showing only the green channel. Given the importance of this figure to the overall conclusions, there needs to be some attempt at quantification here and I’d also like to see controls shown to ensure no ‘bleed through’ between green and red channels.
The serial infection experiment shown in Figure 7 is interesting but no untreated control is shown to show the effect of PI treatment. As such, it feels a bit out of place and should maybe be moved into the supplementary material.
Reviewer 2 Report
The manuscript entitled “Anti-Retroviral Protease Inhibitors Regulate Human Papillomavirus 16 Infection of Primary Oral and Cervical Epithelium” by Dr. Meyers and collaborators reports the development of in vitro model of HPV infection of highly active anti-retroviral therapy naïve primary human gingiva and cervical epithelium. This innovative study shows the potential of this in vitro model to reproduce an in vitro human epithelium capable of replicating the complete HPV16 life cycle. The study also highlights the important notion that highly active anti-retroviral therapy may be a co-factor of HPV infection and HPV-driven tumors onset
Strengths
The information is presented in a comprehensive/clear way and covers a lot.
The literature is adequate
Innovative work
Limitations
Figures are not demonstrative; they are almost unreadable
I have only few comments/suggestions.
Abstract
HIV should be indicated as human immunodeficiency virus (HIV)
Introduction
Page 1 line 38 HAART should be indicated as highly active anti-retroviral therapy (HAART)
Page 2 Line 73 As correctly indicated by the authors, “Oncogenic high-risk HPV16 is responsible for more than 60% of oropharyngeal carcinoma, and ~90% of tonsillar carcinomas.” However, other important HPV16-driven tumors should be briefly mentioned. Indeed, penile/anus cancers (PMID: 21543996) as well as vulvar cancer (PMID: 32266002), have been found to be driven by oncogenic HPV infection. For completeness, the tumors mentioned above, along with references, should be quoted in this section.
Methods
More information about the origin of the virus stock should be given
Page 16 line 525, please indicate the number of patients enrolled for the study
Results
Fig 5, panel A. It is not clear why there is a decrease in intensity of L1 capsid complexes staining at 24hrs and 72hrs
Discussion
Page 15 line 456 Are there information about the relationship between ART and other HPV-driven caners such as vulvar and/or penile cancers?
Page 15 line 447 “…establishment of oral disease.” Please include supporting references
Figures
Several figures, including figure 1, 4 and 6, are too small, almost unreadable. The authors should enlarge the fonts/words, or move some of those panels to the supplemental material.
Figure 2 panel C, BrdU column, there is probably a problem during the uploading of this panel, the figure shows a vertical line